# Cytosolic retention of HtrA2 during mitochondrial protein import stress triggers the DELE1-HRI pathway
Paul Y. Bi[1], Samuel A. Killackey[1], Linus Schweizer[1], Damien Arnoult [2], Dana J. Philpott [1,3] & Stephen E. Girardin [1,3] ✉

Mitochondrial stress inducers such as carbonyl cyanide m-chlorophenyl hydrazone (CCCP) and oligomycin trigger the DELE1-HRI branch of the integrated stress response (ISR) pathway. Previous studies performed using epitope-tagged DELE1 showed that these stresses induced the cleavage of DELE1 to DELE1-S, which stimulates HRI. Here, we report that mitochondrial protein import stress (MPIS) is an overarching stress that triggers the DELE1-HRI pathway, and that endogenous DELE1 could be cleaved into two forms, DELE1-S and DELE1-VS, the latter accumulating only upon non-depolarizing MPIS. Surprisingly, while the mitochondrial protease OMA1 was crucial for DELE1 cleavage in HeLa cells, it was dispensable in HEK293T cells, suggesting that multiple proteases may be involved in DELE1 cleavage. In support, we identified a role for the mitochondrial protease, HtrA2, in mediating DELE1 cleavage into DELE1-VS, and showed that a Parkinson's disease (PD)-associated HtrA2 mutant displayed reduced DELE1 processing ability, suggesting a novel mechanism linking PD pathogenesis to mitochondrial stress. Our data further suggest that DELE1 is likely cleaved into DELE1-S in the cytosol, while the DELE1-VS form might be generated during halted translocation into mitochondria. Together, this study identifies MPIS as the overarching stress detected by DELE1 and identifies a novel role for HtrA2 in DELE1 processing.

Mitochondria are double membrane organelles that are crucial for maintaining cellular homeostasis. Despite containing a separate genome along with the machinery for protein synthesis, the vast majority of mitochondrial proteins are encoded in the nucleus and require active import into mitochondria following translation in the cytosol[1]. The processes of mitochondrial protein import are critical for the normal functions of the mitochondrial system and, as such, are tightly regulated by the cell[1]. Mitochondrial proteins translated in the cytosol enter the mitochondria through the translocase of outer membrane (TOM) complex and are targeted to their final mitochondrial location, which is determined by specific regions of the protein, such as N-terminal mitochondrial targeting sequences (MTS) or internal targeting sequences[1]. In most cases, the MTS is removed by resident proteases, resulting in a mature mitochondrial protein. The import of proteins across the translocase complexes is energy-consuming and dependent on factors such as the establishment of the mitochondrial inner membrane potential ($\Delta\Psi$m), ATP availability and the mitochondrial chaperone system.

Defective mitochondrial protein import not only interferes with the function and fitness of the mitochondrial network, but it additionally leads to the cytosolic accumulation of unimported proteins, which can aggregate and induce proteotoxicity in the cytosol[2,3]. Multiple stress response pathways have evolved to specifically detect and respond to mitochondrial protein import stress (MPIS), which lead to either the repair or clearance of the mitochondria. In budding yeast, MPIS caused by the clogging of the TOM complex can induce the mitochondrial compromised protein import response (mitoCPR), which actively removes the stuck presequence-containing proteins[4]. In nematodes, the mitochondrial unfolded protein response (UPR^mt) triggered by protein aggregation in the inter membrane space (IMS) and the matrix leads to up-regulation of genes that encode mitochondrial chaperones such as *dnj-10* and *hsp-6*[5]. Previous work from

[1]Department of Laboratory Medicine and Pathobiology, University of Toronto, Toronto, ON M5S 1A8, Canada. [2]INSERM U1197, Hôpital Paul Brousse, Bâtiment Lavoisier, Villejuif, Cedex 94807, France. [3]Department of Immunology, University of Toronto, Toronto, ON M5S 1A8, Canada. ✉e-mail: stephen.girardin@utoronto.ca

our lab showed that prolonged MPIS in mammalian cells is the major driver for the mitochondria to be targeted for mitophagy, which is the sequential labeling and removal of dysfunctional mitochondria by the autophagy machinery[6].

Cytosolic accumulation and aggregation of mitochondrial precursors can trigger the misfolding of other additional proteins in the cytosol[7]. Recent studies from our lab described the cytosolic unfolded protein response (cUPR), a response that involves the sensing of cytosolic protein aggregates by heme-regulated inhibitor (HRI), which is one of the key kinases of the integrated stress response (ISR)[8]. The ISR is an evolutionarily conserved pathway that can be activated by a variety of cellular stresses, such as viral infection and amino acid starvation[9]. Upon activation of the ISR, specific stress response proteins such as activating transcription factor 4 (ATF4) are upregulated, while general cap-dependent mRNA translation is inhibited, thereby resulting in a reduction of the proteotoxic stress[9]. Recently, several reports linked the activation of the HRI-directed ISR to mitochondrial stress, through a little-known mitochondrial protein named DELE1[10,11]. Initial studies suggested that when mitochondria were challenged with stress inducers such as the ΔΨm uncoupler carbonyl cyanide m-chlorophenyl hydrazone (CCCP) and the ATP synthase inhibitor oligomycin, DELE1 was cleaved by the IMS protease OMA1 to form a shorter fragment named DELE1-S, which subsequently relocated into the cytosol[10,11]. DELE1-S then interacts with HRI to initiate the ISR, leading to the up-regulation of stress response factors such as ATF4 and CHOP[10,11]. However, due to the difference in the mechanisms of action of CCCP and oligomycin, the nature of the overarching mitochondrial stress triggering the DELE1-HRI pathway remained unclear. A subsequent report suggested that when the mitochondrial protein import machinery was inhibited by CCCP or oligomycin, matrix-targeting DELE1 remained stuck in the translocase complex upon import[12]. The cleavage of DELE1 could then occur and MPIS was identified as the overarching stress that triggers the ISR[12]. One limitation of these studies was the reliance on detecting a C-terminally tagged version of DELE1, whether in replacement of the endogenous locus or following overexpression, due at that time to the lack of antibodies that could recognize endogenous DELE1[10–12]. However, how this pathway would function in a completely endogenous cellular system remained unexplored. Here, we elucidated the molecular underpinnings of DELE1-HRI induced ISR based on the endogenous detection of DELE1. We report that MPIS is the overarching stress that induces cleavage of endogenous DELE1 and triggers HRI-dependent ISR, and provide evidence that the processing of DELE1 into DELE1-S likely occurs in the cytosol before import, arguing that the cleaved protein is not retro-translocated from the IMS to the cytosol as previously proposed. We also discovered a novel cleavage of DELE1 that likely occurs only when the protein is stuck in the mitochondrial import machinery. Surprisingly, we found that the importance of OMA1 for the cleavage of DELE1 varied in different cell line models, suggesting that DELE1 cleavage might be promiscuous. We further identified HtrA2, a mitochondrial protease that was previously linked to mitochondrial fitness and development of neurodegenerative diseases, as a new DELE1 protease that appeared to cleave preferentially DELE1 at a new site that we identified. Interestingly, HtrA2-dependent cleavage of DELE1 was blunted in the G399S HtrA2 variant associated with Parkinson's disease (PD), suggesting the existence of an unexpected link between PD pathogenesis and the DELE1-HRI branch of the ISR.

## Results
### The DELE1-HRI pathway activates the ISR in response to MPIS
To validate the new anti-DELE1 antibody that we obtained, we first studied the activation of mitochondrial stress-induced ISR in HEK293T cells. When cells were treated with CCCP for 4 h to depolarize the inner mitochondrial membrane, we observed a robust induction of ATF4, as well as the formation of a DELE1 band at around 40 kDa, consistent with the 40.29 kDa predicted molecular weight of DELE1-S (Fig. 1a). However, our antibody against endogenous DELE1 did not detect the 55 kDa full length DELE1 long form (DELE1-L) in baseline conditions in HEK293T cells, which could

be explained by the rapid import and degradation of endogenous DELE1 upon entering the mitochondria[12]. Interestingly, the antibody was able to detect DELE1-L at baseline in lysates of HeLa cells (Supplementary Fig. 1a), again supporting the notion that the difficulty in detecting the DELE1-L form is not an intrinsic issue with the antibody but rather reflects the fact that this proform does not accumulate in HEK293T, likely because of a rapid turnover once the protein enters mitochondria. In HeLa cells, we speculate that the protein either displays a slower turnover or is expressed at higher levels, allowing to capture enough of the proform to be detected by western blot. When HEK293T cells were treated with oligomycin, an ATP synthesis inhibitor that blocks the proton channel of the ATP synthase, in addition to DELE1-S, an even shorter DELE1 band appeared around 37 kDa, and we coined this shorter form as DELE1-very short or "DELE1-VS" (Fig. 1b). DELE1-VS was also observed in HeLa cells treated with oligomycin (Supplementary Fig. 1a). Trans-Resveratrol, another ATP synthesis inhibitor that targets the rotary mechanism of the ATP synthase[13], also induced both DELE1-S and DELE1-VS similarly to oligomycin (Supplementary Fig. 1b). We then confirmed that DELE1-S and DELE1-VS were not generated from post-lysis degradation, by lysing HEK293T cells in boiling hot Laemmli buffer during collection. Indeed, we detected similar levels of DELE1-S and DELE1-VS regardless of the method of lysis (Fig. 1b). Since the induction of ATF4 and cleavage of DELE1 are our main readouts for ISR activation, we aimed to confirm, in our cellular system and with our new anti-DELE1 antibody, the previously reported involvement of HRI and DELE1 in CCCP-induced ISR by knocking down protein expression using shRNA. When HEK293T cells were challenged with CCCP, we observed the induction of ATF4 in shRNA control (SC) cells (Supplementary Fig. 1c) while the induction was dampened in cells deficient in HRI or DELE1, thus confirming the crucial role of HRI and DELE1 in the ISR following depolarizing mitochondrial stress. We also noted that the formation of DELE1-S was not HRI dependent, which supports previous work positioning HRI downstream of DELE1 (Supplementary Fig. 1c)[10,11]. However, interestingly, we also noted that ATF4 induction in response to mitochondrial stressors was only marginally affected by HRI or DELE1 silencing in HeLa cells, suggesting the existence of additional pathways in HeLa cells that can relay mitochondrial stress to the ISR (Supplementary Fig. 1d). In addition, HRI knockdown in HeLa cells greatly reduced the amount of DELE1-S formed without largely affecting the level of DELE1-L (Supplementary Fig. 1d, e), which suggested the existence of an uncharacterized HRI-related regulatory feedback loop that affects DELE1 cleavage. Since mitochondrial depolarization induces MPIS, subsequently leading to mitophagy as shown in our previous study[6], we hypothesized that MPIS could be the overarching stress that triggers the DELE1-HRI directed ISR. To test this hypothesis, we first treated shHRI and shDELE1 HEK293T cells with additional mitochondrial stressors that induce MPIS. Whereas CCCP, oligomycin plus antimycin (O + A) and oligomycin alone trigger general protein import inhibition, MitoBlock-6 (MB-6) specifically blocks protein import into the intermembrane space (IMS) by blocking the Mia40-Erv1 pathway[14]. In SC cells, all the tested drugs activated the ISR, indicated by ATF4 induction and DELE1 cleavage, whereas in cells deficient in DELE1 or HRI, the induction of ATF4 was greatly reduced (Fig. 1c, d). We then tested an inhibitor named MitoBlock-10 (MB-10), which inhibits the import of proteins into the mitochondrial matrix by blocking the pre-sequence translocase-associated motor that works in combination with the TIM23 complex[15]. Similar to MB-6, MB-10 induced DELE1 cleavage and ATF4, while the latter was diminished in the absence of HRI/DELE1 (Fig. 1e).

We next tested if proteasome inhibition could also induce the HRI-DELE1 axis. Indeed, previous studies, including our own work[16–18], have shown that proteasome inhibition is a potent inducer of HRI signaling, and recent evidence suggests that the removal of misfolded proteins that accumulate at the site of mitochondrial protein import may be essential to prevent clogging of the mitochondrial import machinery, thus suggesting that MG132-dependent activation of HRI might be at least in part dependent on DELE1[16–18]. In support, we found that high dose of MG-132 treatment (50 μM) was sufficient to induce accumulation of DELE1-S and

**Fig. 1 | The DELE1-HRI pathway activates the ISR in response to MPIS. a** WT HEK293T cells were treated with DMSO or CCCP (20 µM) for 4 h and whole-cell lysates were analyzed by Western blot (WB). Molecular weight markers are provided on the left of the blots in kilodalton (kDa). **b** WT HEK293T cells were treated with DMSO, CCCP (20 µM) or oligomycin (10 µM) for 4 h, and then collected in either cold PBS or in boiling hot Laemmli buffer with SDS. Whole-cell lysates were analyzed by WB. **c** shRNA control (SC) and HRI A shRNA knockdown (shHRI) HEK293T cells were treated with DMSO or a panel of mitochondrial stress inducers for 4 h and whole-cell lysates were analyzed by WB. O + A is oligomycin plus antimycin co-treatment. The concentration of drugs used are: CCCP 20 µM, MB-6 300 µM, oligomycin 100 µM, antimycin 1 µM. **d** SC and shDELE1 A HEK293T cells were treated with DMSO or the same panel of mitochondrial stress inducers as in (**c**) for 4 h and whole-cell lysates were analyzed by Western WB. **e** SC, shHRI A and shDELE1 A HEK293T cells were treated with DMSO, oligomycin (10 µM) or MB-10 (100 µM) for 4 h and whole-cell lysates were analyzed by WB. **f** WT HEK293T cells were treated with DMSO, CCCP (20 µM), oligomycin (10 µM) or MG-132 (50 µM) for 4 h and whole-cell lysates were analyzed by WB. **g** WT HEK293T cells were transfected with or without 0.4 µg over-expression (OE) plasmid of DELE1-HA overnight and treated with DMSO or CCCP (20 µM) for 4 h. Whole-cell lysates were analyzed by WB. **h** WT HEK293T cells were treated with DMSO, CCCP (20 µM) or oligomycin (10 µM) for 4 h, or transfected with an increasing amount of NLRX1-FLAG OE plasmid overnight. Whole-cell lysates were analyzed by WB.

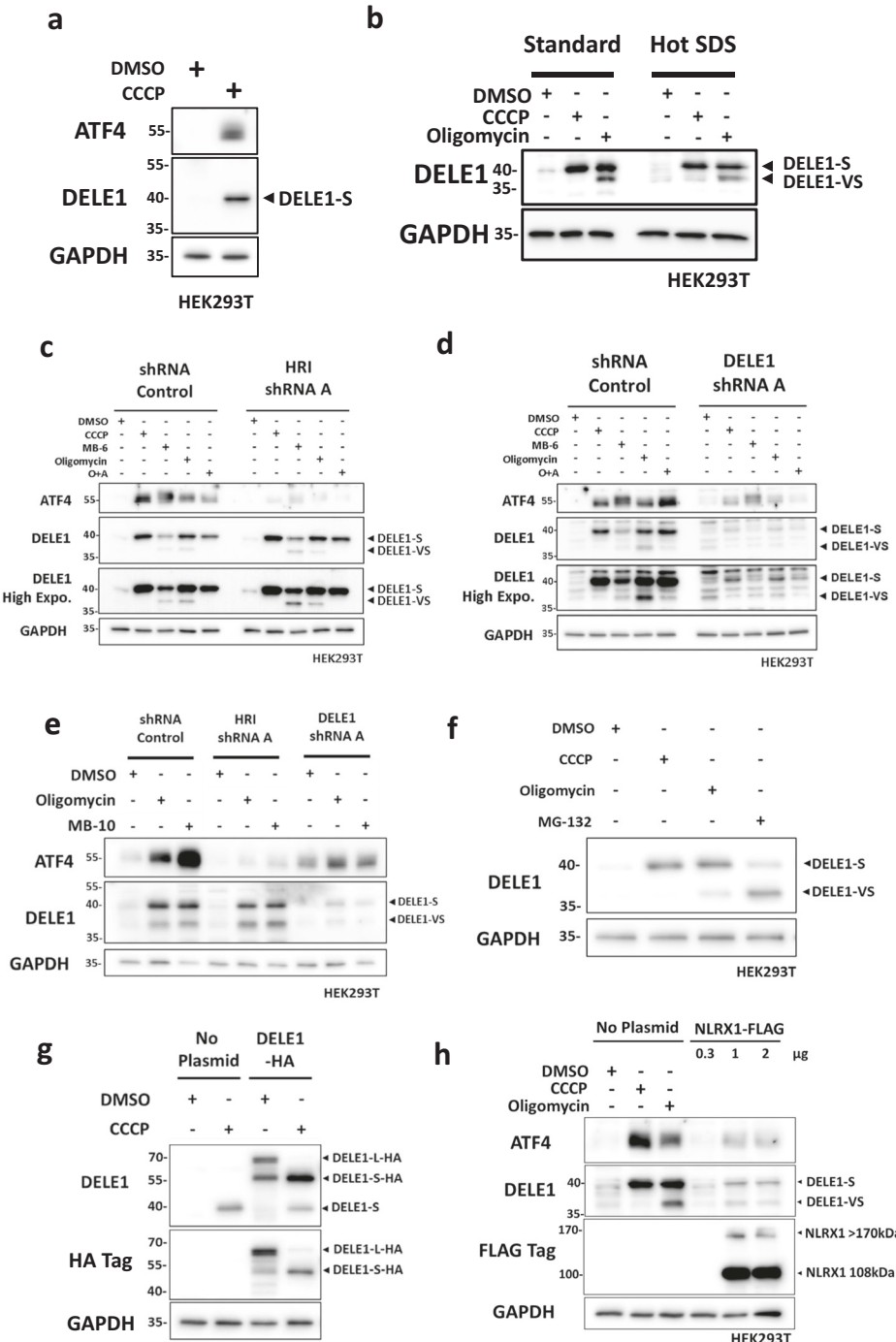

DELE1-VS (Fig. 1f), even without addition of chemical MPIS inducers. However, the accumulation of DELE1 cleavage products after MG-132 treatment could also be the result of disruption in proteosome-directed degradation of constitutively produced DELE1-S and DELE1-VS. To study the potential involvement of the proteasome in the clearance of DELE1 cleavage products, HEK293T cells were first treated with CCCP or oligomycin to induce cleavage of DELE1 and then cells were cultured in normal media to recover from MPIS (Supplementary Fig. 1f, g). A small dose of MG132 (1 µM) which by itself was insufficient to induce DELE1 cleavage was added to inhibit the proteasome. In the CCCP recovery experiments (Supplementary Fig. 1f), addition of MG132 led to elevated levels of DELE1-S which took longer to recover to baseline, suggesting that the proteasome is playing a role in the degradation of DELE1-S. In the oligomycin recovery

experiments (Supplementary Fig. 1g), MG132 treatment resulted in elevated levels of both DELE1-S and DELE1-VS, thus confirming the results with CCCP and indicating that the proteasome is indeed involved in the removal of both DELE1-S and -VS following MPIS. To specifically study the effect of cytosolic accumulation of mitochondrial presequences, we induced MPIS simply by over-expressing mitochondrial proteins to overwhelm the import machinery. Using HEK293T cells that transiently over-express DELE1 with a C-terminal linker region followed by three HA tags on the C-terminus (DELE1-L-HA; expected molecular weight ~70 kDa), a fraction of DELE1-L-HA was spontaneously cleaved to DELE1-S-HA upon over-expression, without the addition of CCCP (Fig. 1g). This suggests that overloading the mitochondrial import machinery with over-expressed DELE1 might be a signal sufficient to induce spontaneous cleavage of DELE1-L into DELE1-S.

**Fig. 2 | Inhibition of the TIM23 complex activates DELE1 signaling. a** si Negative control, siTOM20, siMIA40, siTIM22 and siTIM23 HEK293T cells were collected 2 days post siRNA transfection and whole-cell lysates were analyzed by WB. **b** si Negative control, siTOM20, siTOM40, siTOMM70, siMIA40, siTIM22 and siTIM23 HEK293T cells were collected 4 days post siRNA transfection and whole-cell lysates were analyzed by WB. WT HEK293T cells treated with DMSO or oligomycin (10 μM) were included as positive control. **c** SC, shHRI A and shDELE1 A HEK293T cells were transfected with no siRNA, negative control siRNA or TIM23 siRNA for 2 days and whole-cell lysates were analyzed by WB.

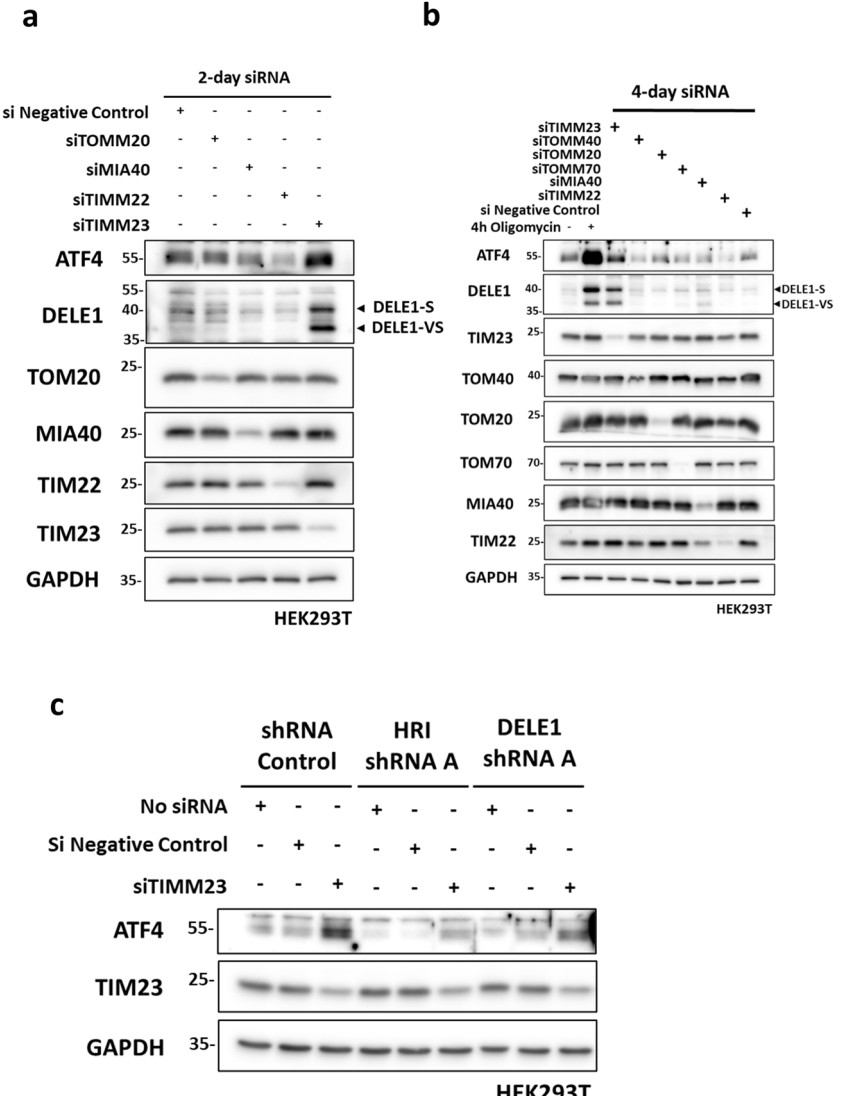

Importantly, our DELE1 antibody was capable of detecting DELE1-L-HA after over-expression, which is in support of the rapid import and turnover of endogenous DELE1 that may explain the inability of detectable DELE1-L in the absence of overexpression in HEK293T cells (see also above). To extend these findings, we asked if other nuclear-encoded, mitochondrial proteins could also lead to the induction of the DELE1-HRI pathway when overexpressed. NLRX1 is another mitochondrial protein imported into the mitochondrial matrix that we have studied in detail previously[19,20]. Over-expression of a C-terminal FLAG-tagged construct of NLRX1 (NLRX1-FLAG) was sufficient to trigger the cleavage of endogenous DELE1 and ATF4 induction (Fig. 1h), whereas the over-expression of a form of NLRX1 that lacks the mitochondrial targeting sequence (ΔMTS-NLRX1-FLAG) did not lead to DELE1 cleavage (Supplementary Fig. 1h), confirming the importance of disrupted mitochondrial protein import on the induction of DELE1 cleavage. Together, these results identify MPIS as the overarching stress that triggers endogenous DELE1 cleavage and activation of the DELE1-HRI axis of the ISR.

## Inhibition of the TIM23 complex activates DELE1 signaling

In order to delineate the molecular requirement for activation of endogenous DELE1 during MPIS, we used siRNAs to silence key components of the various mitochondrial protein import pathways, in order to directly induce MPIS. We targeted TOM20 for general protein import across the outer mitochondrial membrane (OMM), Mia40 for import to the IMS, TIM22 for import to the inner membrane (IMM) and TIM23 for the import to the matrix[1]. Among these conditions, only the knockdown of TIM23 induced a strong ATF4 response and DELE1 cleavage, thus further linking MPIS to the activation of ISR (Fig. 2a). We obtained similar results when siRNA-mediated silencing was maintained for 4 days (Fig. 2b). In the latter condition, we additionally included siTOM40, siTOM70 and again, inhibition of protein import to the matrix by TIM23 knockdown remained as the sole condition able to trigger robust ATF4 induction and DELE1 cleavage (Fig. 2b). We also simultaneously silenced expression of TOMM20 and TOMM70, since both proteins contribute to the import through the TOM. Once again, only siTIMM23 but not siTOMM20/siTOMM70 was able to trigger DELE1 cleavage (Supplementary Fig. 2). To confirm the roles of HRI and DELE1 during TIM23 knockdown-induced ISR, we transfected siRNA targeting TIM23 in SC, shHRI and shDELE1 cells (Fig. 2c). As expected, ATF4 induced by TIM23 knockdown was reduced in the absence of the DELE1-HRI axis. Together, these results suggest that the DELE1-HRI pathway relays the stress triggered by the inhibition of mitochondrial protein import to the ISR and, more specifically, identify the inhibition of the TIM23 translocase of the IMM as the key event inducing the DELE1-HRI pathway during MPIS.

## C-terminal processing of DELE1 after non-depolarizing MPIS

While testing different mitochondrial stress inducers, we noticed that certain treatments lead to the formation of DELE1-S, such as CCCP and O + A, while oligomycin or the two MitoBlock compounds lead to the formation of DELE1-S and the shorter DELE1-VS (see Fig. 1c−e and Supplementary Fig. 1b). Importantly, this shorter band indeed corresponded to a DELE1 form since it was blunted in shDELE1 cells (see Fig. 1d and Supplementary Fig. 1a). Based on the differing mechanisms of action between these two treatment categories, we suspected that the mitochondrial membrane potential (ΔΨm) might be a key factor for DELE1-VS formation, since CCCP and O + A lead to the depolarization of ΔΨm, whereas oligomycin hyperpolarizes ΔΨm, and MB-6 and MB-10 do not alter ΔΨm. We first aimed to confirm that these treatments indeed affected or not the ΔΨm as expected. To do so, we utilized a fluorescence-based tetramethylrhodamine, ethyl ester, perchlorate (TMRE) assay to monitor the change in ΔΨm in cells that were treated with our panel of drugs (Fig. 3a). Similar to the non-treated cells, oligomycin-, MB-6- and MB-10-treated cells displayed mitochondria that were labeled by TMRE, indicating the maintenance of the mitochondrial inner membrane potential. In contrast, CCCP- and O + A-treated cells showed minimal or no fluorescence, as a result of mitochondrial depolarization. We then aimed to determine in what region of the DELE1 protein does the cleavage occur to generate the DELE1-VS form. According the relative apparent molecular weights of the DELE1-S and DELE1-VS forms, we speculated that, in principle, the "VS" cleavage site could occur at three possible locations (Fig. 3b). To distinguish between these options and assess how the DELE1-VS fragment is generated, we again utilized the DELE1-HA overexpression plasmid. As we noted above, the C-terminus of the DELE1-L-HA protein expressed from this plasmid is tagged with an additional ~12 kDa peptide linker sequence and 3x HA-tags, thereby allowing easy discrimination from the endogenous protein. When DELE1-HA was over-expressed overnight, we again observed the spontaneous cleavage of the ~70 kDa DELE1-L-HA into ~55 kDa DELE1-S-HA in the absence of treatment (Fig. 3c) and, as expected for a "S" cleavage site towards the N-terminal end of the protein, the DELE1-S-HA form was detected by both the anti-DELE1 and anti-HA antibodies. Higher exposures of our DELE1 blots indicated that the endogenous ~37 kDa DELE1-VS was also formed spontaneously at baseline following over-expression, and was strongly induced after oligomycin treatment but not CCCP treatment. Interestingly, the anti-HA blot failed to detect DELE1-VS, which is in support of a C-terminal cleavage that cuts off the HA tag, and suggests that the "VS" cleavage site is likely occurring as shown as options 2 or 3 in Fig. 3b. To further distinguish between options 2 and 3, it is indispensable to visualize the remaining C-terminal fragment generated following induction of the "VS" cleavage. Indeed, option 2 suggests that the "VS" cleavage site is ~3 kDa from the C-terminal end (since endogenous DELE1-S is ~40 kDa and endogenous DELE1-VS is ~37 kDa), while option 3 would require a "VS" cleavage site ~18 kDa from the C-terminal end (~55 kDa for DELE1-L minus ~37 kDa of DELE1-VS). Since cleaved protein fragments can be rapidly degraded by the proteosome system, we decide to co-treat cells with MPIS inducers either alone or in combination with the proteosome inhibitor MG-132, in order to minimize the degradation of cleaved DELE1 fragments. Interestingly, co-treatment of DELE1-L-HA over-expressing cells with oligomycin plus MG-132 resulted in the detection by our anti-HA antibody of a band at ~20 kDa (Fig. 3d), which is compatible with a "VS" cleavage at 3−5 kDa from the C-terminal end of DELE1 plus ~15 kDa corresponding to the linker region plus 3xHA tag. Thus, we concluded that the "VS" cleavage site occurs towards the C-terminal end of DELE1, resulting in the removal of a 3−5 kDa fragment (option 2, Fig. 3b). In order to further narrow down the region of DELE1-VS cleavage site, we generated a series of truncated DELE1 mutants, which each lacked 10 amino acids towards the c-terminal end (Δ446-456 DELE1, Δ456-466 DELE1, Δ449-468 DELE1 and Δ466-486 DELE1). When we over-expressed these mutants in HEK293T cells, compared to WT DELE1, we observed loss in the level of DELE1-VS following oligomycin in cells with Δ456-466 DELE1 without affecting the generation of DELE1-S (Fig. 3e). The remaining

detectable DELE1-VS likely came from endogenous DELE1 proteins present in these cells. As expected, the C-terminal fragment induced by DELE1-VS cleavage was also missing in Δ456-466 DELE1 expressing cells that were treated with MG-132 and oligomycin (Fig. 3e). In agreement with the results from Δ456-466 DELE1, the reduction of DELE1-VS and C-terminal fragment were also observed in cells expressing Δ449-468 DELE1(Fig. 3e). When the proposed cleavage site of DELE1-S and identified region of DELE1-VS cleavage were mapped using the 3D structure of DELE1 predicted by Alphafold[21], both were found next to or within loop-like regions that lack a defined structural motif (Fig. 3f). Interestingly, by aligning the peptide sequences of DELE1 from multiple species, amino acids in position 456-466 were found to be generally conserved only in mammals (*Homo sapiens, Mus musculus, Bos taurus and Felis catus*) but not in other animal groups such as birds (*Gallus gallus*), reptiles (*Xenopus tropicalis*) and fish (*Danio rerio*) (Fig. 3g). We then went on to mutate individual residues within the 10-amino acid long region in order to find the specific residue(s) responsible for the DELE1-VS cleavage. However, unfortunately we were not able to identify any single amino acid substitution that could largely abolish the DELE1-VS cleavage following oligomycin, suggesting that the cleavage site to generate the VS fragment was likely promiscuous and did not require a very strict amino acid sequence. We next asked what was the potential physiological function of DELE1-VS. We suspected that DELE1-VS would function similarly to DELE1-S, since most of the HRI-interacting tetratricopeptide repeat (TPR) domains are still present in DELE1-VS[11]. In order to study the potential interaction between HRI and DELE1-VS, we utilized co-immunoprecipitation assay in HEK293T cells over-expresseing WT DELE1-HA and HRI-Myc and then induced DELE1-VS cleavage by oligomycin (Fig. 3h). DELE1-S, DELE1-L and DELE-VS were all pulled down by a Myc-tag antibody, suggesting that all three forms of DELE1 were capable of interacting with HRI (Fig. 3h). In summary, our data support the notion that, under non-depolarizing MPIS, DELE1 is additionally processed at a "VS" cleavage site located within a region of AA456 to AA466, and that the resulting DELE1-VS was capable of interacting with HRI similarly to DELE1-S.

## The DELE1-HRI mediated ISR senses and responds to MPIS independently from mitophagy

Since we have previously shown that MPIS is a common, underlying trigger of mitophagy, we next asked whether mitophagy and the ISR pathway are mechanistically related. Using the same shRNA approach as above, we first generated cells deficient in NLRX1, RRBP1 and PINK1 (Supplementary Fig. 3a, b for knockdown validation). When we evaluated the protein level of ATF4 and DELE1-S after CCCP treatment in these knockdown cell lines, we found no difference compared to the SC cells, suggesting that these mitophagy proteins were not involved in the induction of the ISR (Supplementary Fig. 3c). Additionally, when the same panel of MPIS inducers used in Fig. 1c was applied to shNLRX1 cells (Supplementary Fig. 3d) and shRRBP1 cells (Supplementary Fig. 3e), again the levels of ATF4, DELE1-S and DELE1-VS were comparable to the levels in SC cells, therefore further confirming that MPIS-induced ISR is not regulated by the mitophagy pathway. We also included WT and ATG16L1 knockout HCT116 cells to further rule out the involvement of mitophagy during ISR (Supplementary Fig. 3f). ATG16L1 is a key regulator of all forms of autophagy and the removal of this gene lead to the lack of autophagosome formation[22], which is further evidenced here by the absence of LC3-I to LC3-II conversion in our ATG16L1⁻/⁻ cells (Supplementary Fig. 3f). When these cells were challenged with CCCP and oligomycin, again we observed similar levels of ATF4 induction and DELE1 cleavage when comparing WT and ATG16L1⁻/⁻ cells. We therefore concluded that the MPIS-ISR is likely not regulated by mitophagy pathways. Conversely, we went on to investigate the potential involvement of MPIS-ISR inducers in the activation of mitophagy. To trigger mitophagy, SC, shHRI and shDELE1 cells were treated with CCCP overnight (Supplementary Fig. 3g), and the recruitment of LC3-II to the heavy membrane fraction of cell lysates was assessed by mitochondrial fractionation. Prolonged incubation with

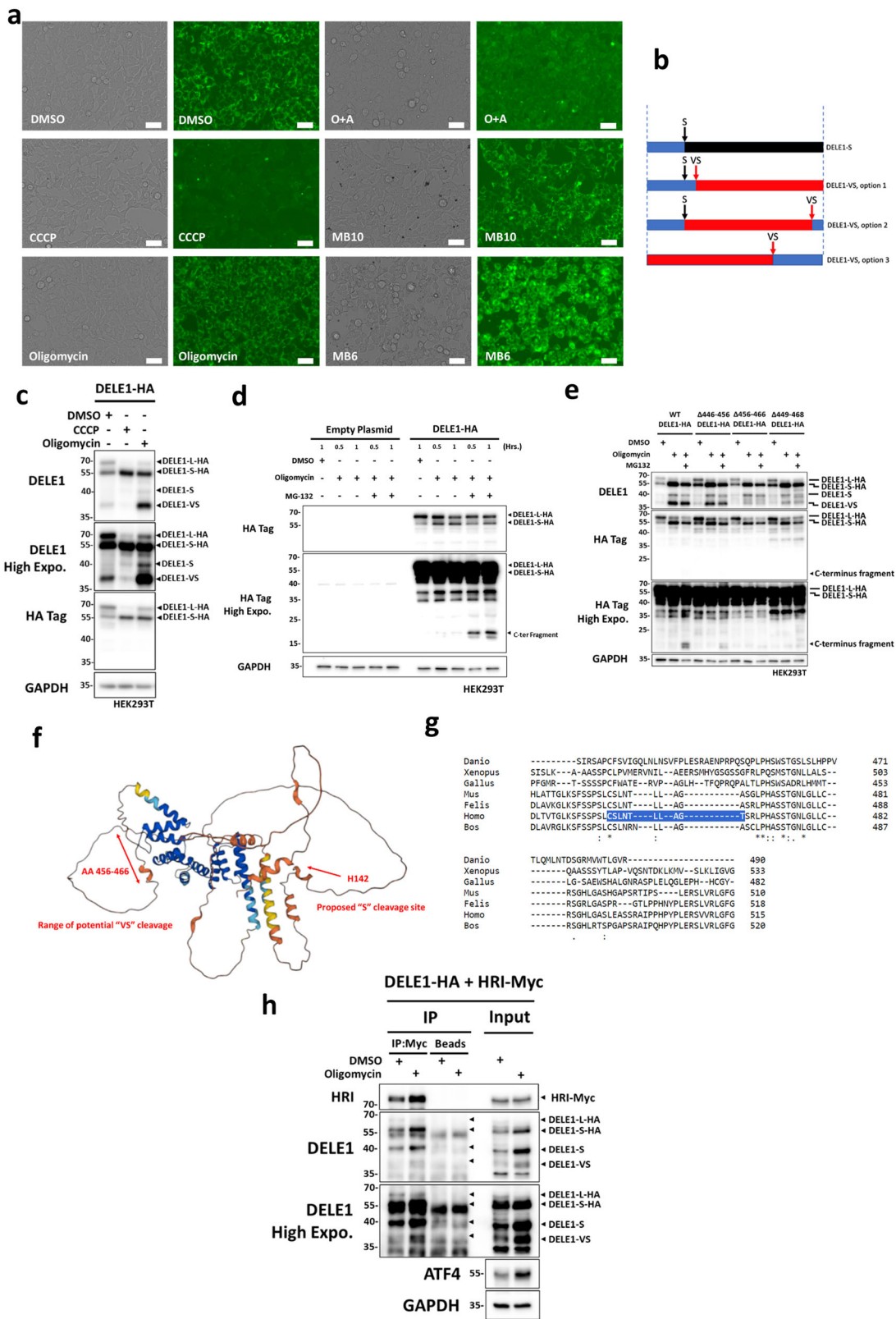

CCCP induced comparable level of LC3-II in the mitochondrial/heavy membrane fraction among the SC and knockdown cells, which indicated that mitophagy progression was not affected when key components of the mitochondrial ISR were silenced. We therefore concluded that mitophagy and the ISR are two independent stress-response pathways that are both activated by MPIS.

## Cell type-dependent cleavage of DELE1 by OMA1 following MPIS

Previous studies have identified OMA1 as the IMS protease that cleaves DELE1 in response to MPIS, while DELE1 is stalled within the translocase complexes[10–12]. After confirming the importance of HRI and DELE1 during MPIS-induced ISR in HEK293T cells, we next shifted our focus to OMA1. We started by transiently knocking down OMA1 using shRNA, and treated

**Fig. 3 | C-terminal processing of DELE1 after non-depolarizing MPIS. a** WT HEK293T cells were treated with DMSO, CCCP (20 μM), oligomycin (10 μM), O + A (10 μM and 1 μM), MB-6 (300 μM) and MB-10(100 μM) for 4 h and analyzed by TMRE staining. Scale bar represents 36 μm. **b** Model of potential DELE1 cleavage sites and cleavage products. From top to bottom: DELE1-S cleavage; An additional N-terminus cleavage to generate DELE1-VS; An additional C-terminus cleavage to generate DELE1-VS; A C-terminus cleavage to generate DELE1-VS without S cleavage. **c** WT HEK293T cells were transfected with DELE1-HA OE plasmid overnight, and then treated with DMSO, CCCP (20 μM) or oligomycin (10 μM) for 4 h. Whole-cell lysates were analyzed by WB. **d** WT HEK293T cells were transfected with DELE1-HA OE plasmid overnight, and then treated with DMSO, oligomycin (10 μM) or oligomycin plus MG-132 (50 μM), for the indicated time. Whole-cell lysates were analyzed by WB. **e** WT HEK293T cells were transfected with WT DELE1-HA, Δ445-456 DELE1-HA, Δ456-466 DELE1-HA and Δ449-468 DELE1-HA OE plasmids overnight, and then treated with DMSO, oligomycin (10 μM) or oligomycin plus MG-132 (50 μM) for 4 h. Whole-cell lysates were analyzed by WB. **f** Alphafold model of DELE1 with the proposed DELE1-S cleavage site and the potential range of DELE1-VS cleavage. **g** Alignment of C-terminus regions of DELE1 from *Homo sapiens*, *Mus musculus*, *Bos taurus*, *Felis catus*, *Gallus gallus*, *Xenopus tropicalis* and *Danio rerio*. The estimated range of DELE1-VS cleavage site (AA 456-466) is highlighted in the sequence of *Homo sapiens*. **h** Myc-tag was immunoprecipitated from lysates of HEK293T cells over-expressing with DELE1-HA and HRI-Myc for overnight and then treated with oligomycin (10 μM) or DMSO for 4 h, and analyzed by WB.

these cells with oligomycin or O + A for 4 h, to monitor ATF4 induction and DELE1 cleavage (Fig. 4a). Unexpectedly, OMA1 knockdown did not prevent the formation of DELE1-S or DELE1-VS after oligomycin treatment, although accumulation of the DELE1-S form induced by O + A co-treatment was slightly reduced in shOMA1 cells compared to the SC, but the intensity of ATF4 upregulation was once again comparable. While OMA1 seemed to play a minimal role at 4 h of treatment, it remained possible that OMA1 could cleave DELE1 relatively early and, by 4 h, the cleavage of DELE1 could be compensated by other proteases in shOMA1 cells. To rule out this possibility, we included time points as early as 15 min in oligomycin (Fig. 4b) and CCCP (Fig. 4c) time curve experiments, in SC and shOMA1 cells. Once again, the results showed that OMA1 was dispensable for DELE1 cleavage in our experimental system. To confirm the knockdown of OMA1 functionally, in addition to the OMA1 blot, we assessed the levels of the protein OPA1, which is a known OMA1 substrate, as OMA1 cleaves full length OPA1 (OPA1-L, a and b) into shorter OPA1 fragments (OPA1-S, c and e) to prevent IMM fusion after mitochondrial stress[23]. As expected, OPA1-L cleavage induced by oligomycin was nearly complete at 4 h in SC cells but at the same time point in shOMA1 cells, the level of OPA1-L was only slightly reduced (Fig. 4b). Similarly, while in CCCP-treated SC cells OPA1-L was almost completely cleaved into OPA1-S within 15 min, such conversion in OMA1 knockdown cells was delayed up to 2 h (Fig. 4c). Furthermore, under baseline condition, shOMA1 cells lacked the OMA1-specific OPA1-S species c and e compared to SC cells[23]. Therefore, all these observations confirmed the knockdown of OMA1 functionally. Of note, in CCCP-treated SC cells we also observed the gradual disappearance of OMA1 during the course of 4 h, which was likely due to the previously reported stress-induced cleavage by OMA1 itself[24,25] and another mitochondrial protease YME1L[26]. To further support our findings, we used a second OMA1 shRNA (named shOMA1 B to distinguish from the shOMA1 A used in experiments presented in Fig. 4a–c). SC, shOMA1 A and shOMA1 B cells were generated and treated with a panel consisting of DMSO, CCCP, oligomycin and O + A (Fig. 4d). Consistent with the data obtained with shOMA1 A, cells deficient in OMA1 targeted by the B clone induced similar level of ATF4, DELE1-S and DELE1-VS as SC cells after challenges with CCCP, O + A and oligomycin. The lack of apparent inhibition of DELE1 cleavage and ATF4 induction following OMA1 silencing was surprising, and suggested that, in HEK293T cells, OMA1 was either not involved in DELE1 cleavage or was not the sole protease responsible for cleaving DELE1 following MPIS. When CCCP (Fig. 4e) and oligomycin (Fig. 4f) time curve experiments were repeated with the additional shOMA1 B cells, we again concluded that there was no apparent difference in the levels of ATF4 induction and DELE1 cleavage between SC and the OMA1 knockdown expressing cells.

In contrast, when we knocked down OMA1 in HeLa cells with the same shRNA constructs, we observed a nearly complete abolishment of DELE1-S and DELE1-VS cleavage following CCCP, oligomycin and O + A induced MPIS (Supplementary Fig. 4a, b). We reasoned that the contrasting effect of OMA1 silencing between these two cell lines could be due to a higher expression of OMA1 in HeLa cells as compared to HEK293T cells. However, we observed similar (if not even lower) levels of OMA1 levels in HeLa cells as compared to HEK293T cells (Supplementary Fig. 4c), and

concluded that the difference in the requirement of OMA1 in DELE1 processing between these two cell lines was not a result of variations in OMA1 expression levels. Despite playing an important role in DELE1 cleavage in HeLa cells, OMA1 was dispensable for the activation of ISR following MPIS, illustrated by the similar ATF4 level in Supplementary Fig. 4a, which is in line with our results above (see Supplementary Fig. 1d) which showed that, in HeLa cells, the DELE1-HRI pathway appeared dispensable for ATF4 induction in response to CCCP or oligomycin treatments. Similar results were obtained in wild type (WT) versus *Oma1*[−/−] mouse embryonic fibroblasts (MEFs), where comparable levels of ATF4 were noted (Supplementary Fig. 4d, e). Of note, we were not able to determine if Oma1 participated in Dele1 cleavage in these MEFs since our antibody did not react with murine Dele1. Altogether, these results suggest that, while playing an important role in HeLa cells, OMA1 is surprisingly dispensable for the cleavage of DELE1 in HEK293T cells, implying that the identity of the DELE1-processing protease(s) involved in mitochondrial ISR pathway is less clear than previously proposed.

## Subcellular location of DELE1 cleavage and distribution

Determining the location of DELE1 cleavage could offer insights into additional proteases that may mediate DELE1 cleavage and activation. Under baseline condition, endogenously-tagged DELE1 was reported to locate to the matrix of the mitochondria[10], where it was subsequently cleared by the LONP1 protease[27]. We confirmed the LONP1-driven DELE1 degradation by showing the accumulation of uncleaved DELE1-L in LONP1 knockdown HEK293T cells under baseline condition (Fig. 5a), suggesting that DELE1-L is normally imported to the mitochondrial matrix. Regarding the location of MPIS-induced DELE1 cleavage, previous reports have suggested that the "S" cleavage of DELE1 occurs where OMA1 resides, in the IMS, when DELE1 is stuck in the protein import machinery as MPIS occur[10–12]. This model also suggests that DELE1-S then retro-translocates into the cytosol upon cleavage to activate HRI and the downstream ISR, through an undefined mechanism. To interrogate the subcellular distribution of DELE1, we first used CCCP to trigger DELE1 cleavage and fractionated the cell lysates into cytosolic and heavy membrane fractions (Fig. 5b). At 30 min post-treatment, DELE1-S accumulation was evident in the cytosol, and the intensity of the cytosolic DELE1-S band gradually increased along the course of 1.5 h. The vast majority of DELE1-S was found in the cytosolic fraction and the level of DELE1-S in the mitochondrial fraction was barely detectable, suggesting a process where DELE1-S is formed rapidly in the cytosol following the cleavage, rather than accumulating in the mitochondria and then translocating to the cytosol. Next, we asked if this preference of location was also seen with other MPIS inducers such as oligomycin (Fig. 5c). Interestingly, when oligomycin induced MPIS without depolarization, DELE1-S and DELE1-VS were found in both cytosolic and mitochondrial fractions, in contrast to the almost exclusive cytosolic distribution of DELE1-S with depolarization by CCCP (Fig. 5c). Since optimal ΔΨm is required for the full import of proteins across the OMM to their final destinations in the IMS[1], IMM or matrix, we suspected that when ΔΨm is hyperpolarized by oligomycin, DELE1 could still partially enter mitochondria through the TOM complex and the cleavage could occur while the protein is partially inserted into the TOM complex, while

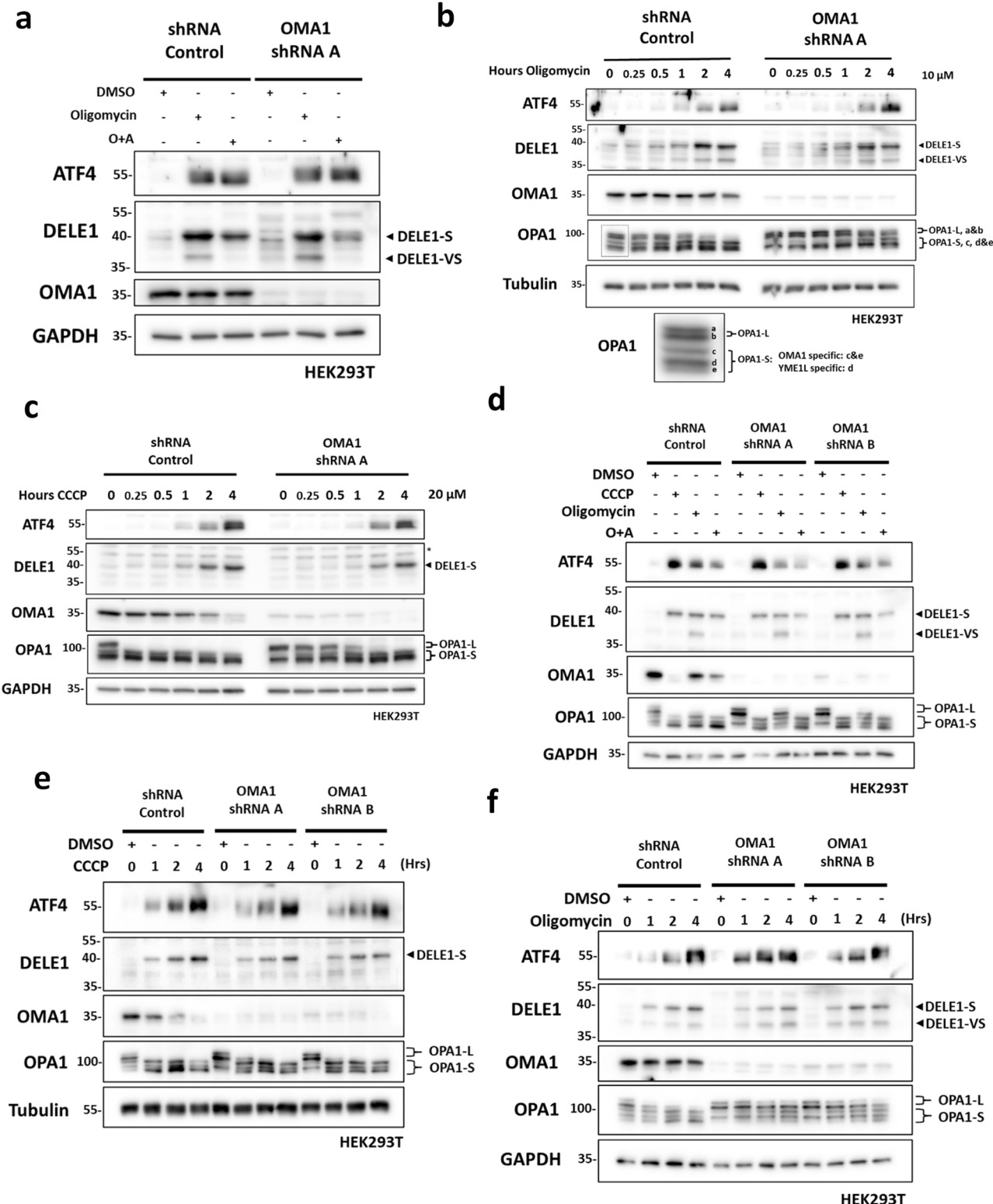

**Fig. 4 | Cell type-dependent cleavage of DELE1 by OMA1 following MPIS. a** SC and shOMA1 A HEK293T cells were treated with DMSO, oligomycin (10 μM) and O + A (10 μM + 1 μM) for 4 h and whole-cell lysates were analyzed by WB. **b** SC and shOMA1 A HEK293T cells were treated with DMSO for 4 h or with oligomycin (10 μM) for 15 min to 4 h and whole-cell lysates were analyzed by WB. Species of OPA1-L and OPA1-S are shown in the cropped-out blot below. **c** SC and shOMA1 A HEK293T cells were treated with DMSO for 4 h or with CCCP (20 μM) for 15 min to 4 h and whole-cell lysates were analyzed by WB. * Indicates non-specific bands.

**d** SC, shOMA1 A and shOMA1 B HEK293T cells were treated with DMSO, CCCP (20 μM), oligomycin (10 μM) and O + A (10 μM and 1 μM) for 4 h, and whole-cell lysates were analyzed by WB. **e** SC, shOMA1 A and shOMA1 B HEK293T cells were treated with DMSO for 4 h or with CCCP (20 μM) for 1 to 4 h and whole-cell lysates were analyzed by WB. **f** SC, shOMA1 A and shOMA1 B HEK293T cells were treated with DMSO for 4 h or with oligomycin (10 μM) for 1 to 4 h and whole-cell lysates were analyzed by WB.

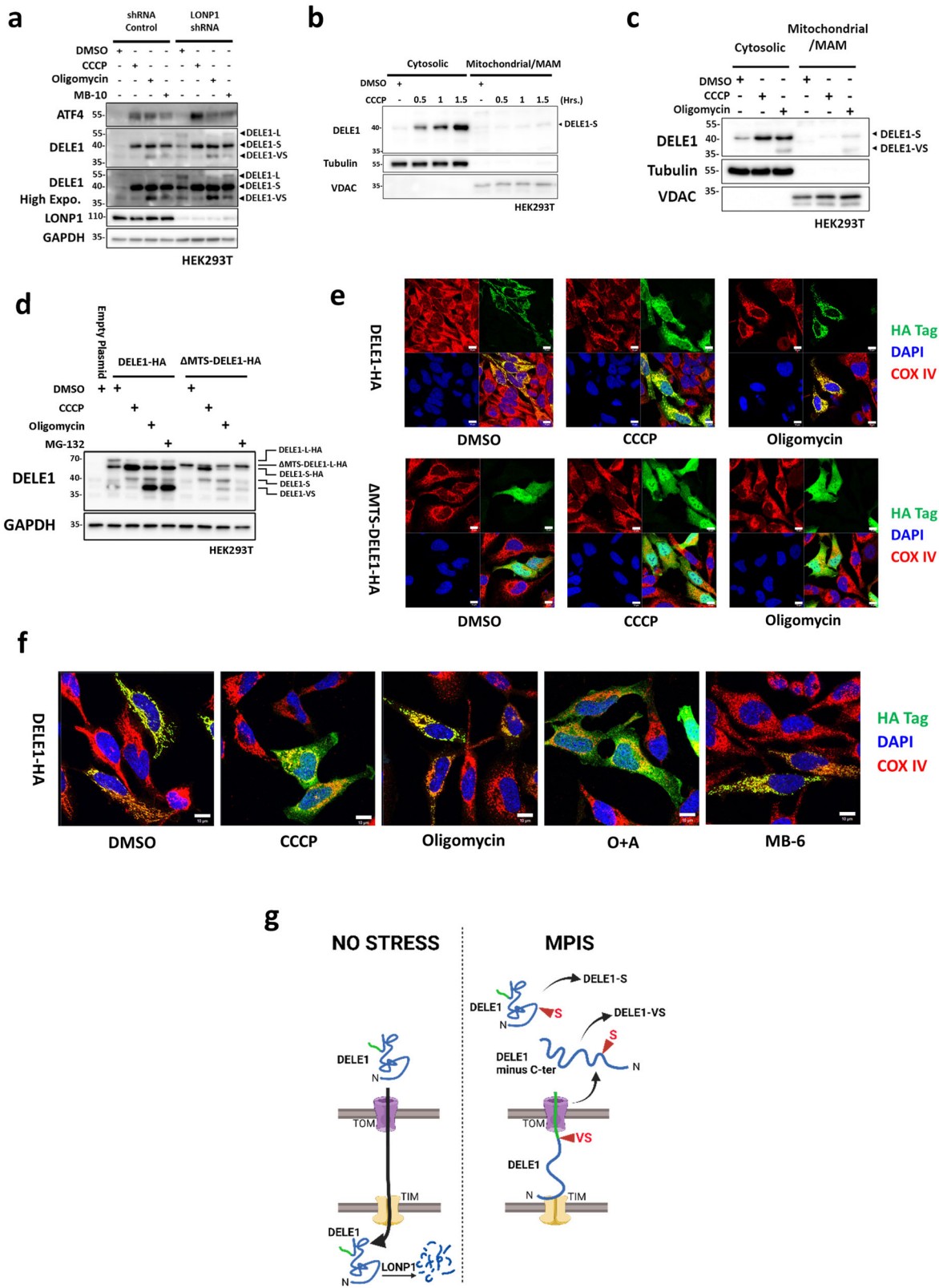

TIM23 complex is inhibited by oligomycin, thereby preventing DELE1 from reaching the matrix. In this scenario, we speculate that the "VS" cleavage might occur while DELE1 is stuck between the TOM and the TIM complexes, which would explain why DELE1-VS is only observed in non-depolarizing MPIS. In contrast, the "S" cleavage would only occur when DELE1 is fully retained in the cytosol, occurring during depolarizing MPIS.

The fact that non-depolarizing MPIS induces the ~37 kDa form that requires both "S" and "VS" cleavage sites (see Fig. 3b, option 2) suggests that, after generation of the "VS" cleavage, DELE1 protein is released from the import complex and accumulates into the cytosol where the "S" cleavage occurs. This model would also explain why the "VS" cleavage site always occurs together with "S" cleavage. Indeed, we could not detect conditions

**Fig. 5 | Subcellular location of DELE1 cleavage and distribution. a** SC and shLONP1 HEK293T cells were treated with DMSO, CCCP (20 μM), oligomycin (10 μM) and MB-10 (100 μM) for 4 h and whole-cell lysates were analyzed by WB. **b** Mitochondrial fractionation of WT HEK293T cells lysates following 1.5 h of DMSO or 30 min, I h and 1.5 h of CCCP (30 μM) and analyzed by WB. **c** Mitochondrial fractionation of WT HEK293T cells lysates following 4 h of DMSO, CCCP (20 μM) or oligomycin (10 μM) and analyzed by WB. **d** WT HEK293T cells were transfected with 0.4 μg of empty plasmid, DELE1-HA OE plasmid, or ΔMTS-DELE1-HA OE plasmid overnight, and then treated with 4 h of DMSO, CCCP (20 μM), oligomycin (10 μM) or MG-132 (50 μM). Whole cell lysates were analyzed by WB. **e** Immunofluorescence (IF) of WT Hela cells over-expressing DELE1-HA or ΔMTS-DELE1-HA for 2 days and then treated with DMSO, CCCP (20 μM) or

oligomycin (10 μM) for 4 h. Scale bar represents 10 μm. **f** Immunofluorescence (IF) of WT Hela cells over-expressing DELE1-HA for 2 days and then treated with DMSO, CCCP (20 μM), oligomycin (10 μM), O + A (10 μM + 1 μM) and MB-6 (300 μM) for 4 h. Scale bar represents 10 μm. **g** Model of DELE1 import and cleavage. Under baseline condition, DELE1 is imported to the matrix and degraded by LONP1. When MPIS occurs, the imported of DELE1 is disrupted. If ΔΨm is depolarized, DELE1 is likely blocked from enter the OMM translocase complex and is subsequently cleaved to form DELE1-S in the cytosol. If ΔΨm is maintained or hyperpolarized, DELE1 might be either stuck in the transport machinery which leads to the formation of DELE1-VS, or cytosolically retained to form DELE1-S. Model is generated from biorender.com.

where a DELE1 form of ~50−52 kDa lacking only the C-terminal end of the protein would be detected.

To provide a more definitive proof that the "S" cleavage of DELE1 occurs in the cytosol, we over-expressed full length DELE1-HA or ΔMTS-DELE1-HA in HEK293T cells, followed by treatment with CCCP, oligomycin and MG-132 (Fig. 5d). Similar to ΔMTS-NLRX1, the complete deletion of N-terminus MTS of DELE1 (previously reported as the first 101 aa[12]) should lead to an exclusively cytosolic protein. The ability for DELE1 to be cleaved independently from mitochondrial import was supported by the observation that under CCCP treatment, ΔMTS-DELE1-HA was also processed into DELE1-S-HA. Strikingly, and in support of the model presented above, the DELE1-VS band that is strongly induced by oligomycin and MG-132 in DELE1-L-HA expressing cells was not found in ΔMTS-DELE1-HA expressing cells, indicating that DELE1-VS requires mitochondrial localization and is possibly formed inside mitochondria, or when DELE1 is engaged into the TOM. To confirm the subcellular location of DELE1-L-HA and ΔMTS-DELE1-HA, we used immunofluorescence (IF) to visualize the distribution of DELE1 using an anti-HA antibody, and compared to the mitochondrial marker COX IV (Fig. 5e). Of note, the distribution of the DELE1-VS form cannot be specifically visualized in these experiments, since the C-terminal cleavage would remove the HA tag of the construct. In baseline conditions, full length DELE1-L-HA co-localized with COX IV, consistent with the reported mitochondrial localization of DELE1. In contrast, ΔMTS-DELE1-HA distributed diffusely throughout the cell, without obvious mitochondrial co-localization (Fig. 5e). After CCCP treatment, the HA epitope from DELE1-L-HA expressing cells also began diffusing into the cytosol, in agreement with the appearance of DELE1-S in the cytosolic fraction of cell lysates after CCCP treatment. Interestingly, in the DELE1-L-HA expressing cells treated with oligomycin, the HA-tagged DELE1 was still associated with the mitochondrial network, which contrasts with reports indicating that oligomycin treatment leads to a diffuse DELE1 localization in the cytosol. While this would support the hypothesis that DELE1 is still at least partially imported into mitochondria after oligomycin treatment, the lack of apparent cytosolic distribution contrasts with the WB of fractionated lysates (see Fig. 5c), which indicated that DELE1-S was found in both the cytosolic and mitochondrial fractions following oligomycin treatment. A possible explanation is that, following the initial "VS" cleavage and release from the TOM-TIM complexes, DELE1 is cleaved on the "S" site immediately at the vicinity of the OMM and accumulates at this site, resulting in an apparent pattern that would remain visually mitochondrial in IF, while it could partition as partially cytosolic and mitochondrial in biochemical fractionation assays. Moreover, the HA signals of ΔMTS-DELE1-HA expressing cells remained broadly distributed in the cytosol after both treatments, as expected (Fig. 5e). Lastly, we analyzed the distribution of HA-tagged proteins in DELE1-L-HA expressing cells treated with O + A or MB-6 (Fig. 5f). Similar to CCCP, mitochondrial depolarization induced by O + A led to DELE1-HA distributing throughout across the cytosol, while the localization of DELE1-HA after MB-6 treatment, which does not affect mitochondrial polarization, remained associated with the mitochondrial network, similar to oligomycin treatment. Overall, these results support a model whereby DELE1 is differentially cleaved by depolarizing vs non-depolarizing MPIS, with a "VS" cleavage occurring at the

mitochondria (likely while the protein is stuck in the entry machinery during non-depolarizing MPIS) and a "S" cleavage occurring either when the protein never engages into the import machinery (depolarizing or non-depolarizing MPIS) or immediately after being sent back to the cytosol after a failed attempt to enter mitochondria and a "VS" cleavage (non-depolarizing MPIS). These findings and the model for DELE1 cleavage during MPIS are summarized in Fig. 5g.

## HTRA2 plays a critical role in DELE1-VS generation

Our data challenge the notion that the cleavage of DELE1 occurs exclusively inside the mitochondria, and that OMA1 is the sole protease responsible. We then investigated the identity of additional proteases that could process DELE1 in response to MPIS. Mitochondrial proteases are a diverse group that contains 25+ members and can be divided into 3 major categories based on their catalytic classes, which are cysteine proteases, serine proteases and metalloproteases[28]. OMA1 is classified as a metalloprotease since it utilizes a $Zinc^{2+}$ ion at the active site to catalyze the hydrolysis of peptide bonds[29]. To narrow down the number of potential protease candidates, we pre-treated cells with protease inhibitors that each targeted specific classes of proteases, while triggering DELE1 cleavage through co-treatment with CCCP and oligomycin (Fig. 6a, b). Interestingly, the serine protease inhibitor 4-(2-Aminoethyl) benzenesulfonyl fluoride hydrochloride (AEBSF), but not the metalloprotease inhibitor o-Phenanthroline (o-Phe), blocked the formation of DELE1 cleavage products following oligomycin and CCCP in a dose-dependent manner (Fig. 6a, b). We also noted that treatment with o-Phe alone was sufficient to induce both forms of short DELE1 without oligomycin treatment, suggesting that general inhibition of metalloproteases might induce some form of constitutive MPIS through unknown mechanisms (Fig. 6a). These results suggested that serine proteases may be able to cleave DELE1. Furthermore, AEBSF did not affect OMA1-directed OPA1 cleavage, as the conversion of OPA1-L to OPA1-S c and e was still evident after treatments with CCCP and oligomycin, thus showing that the inhibitory effect of AEBSF on DELE1 cleavage was not through OMA1 inhibition. Together, these results suggest that in HEK293T cells, the proteases that process DELE1 in response to MPIS are likely to be a serine protease.

Among the mitochondrial serine proteases, high-temperature-required protein A2 (HtrA2) stood out as a potential candidate. HtrA2 was previously identified to play a key role in mitochondrial health and homeostasis[30]. Furthermore, HtrA2 was linked to neurodegenerative diseases, as loss-of-function mutations in *HtrA2* lead to the development of Parkinson-like phenotypes in a mouse model[31,32] and point mutations in *HTRA2* gene in humans were identified in patients with a family history of PD[31,33–35]. Under normal circumstances, HtrA2 is synthesized as a ~ 55 kDa preprotein, which is then imported and cleaved to a ~ 36 kDa mature form in the IMS[36]. Under conditions of apoptotic stress, HtrA2 was shown to translocate to the cytosol as a danger signal and to facilitate the activation of caspases[37,38]. Since our previous results indicated that the cleavage of DELE1 could potentially occur in and out of the mitochondria depending on the nature of the stress, we were interested in HtrA2 serving as a danger signal that can function outside of the mitochondria once import is affected. CCCP-treated cell lysates were fractionated into cytosolic and

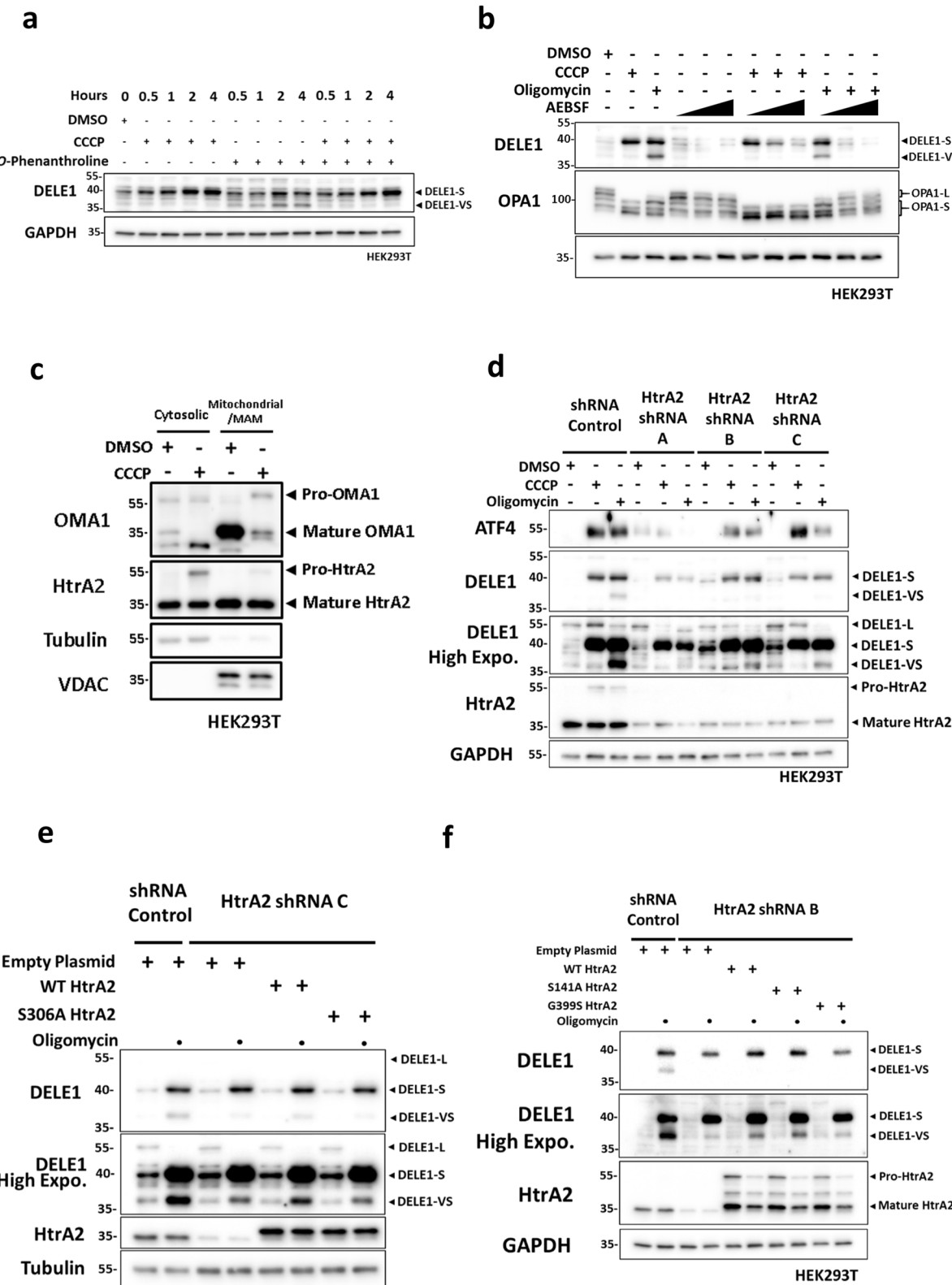

**Fig. 6 | HTRA2 plays a critical role in DELE1-VS generation. a** WT HEK293T cells were treated with DMSO, CCCP (20 μM), *o*-Phenanthroline (500 μM) or a combination of CCCP and *o*-Phenanthroline for the indicated time. Whole-cell lysates were analyzed by WB. **b** WT HEK293T cells were treated with DMSO, oligomycin (10 μM) or increasing doses of AEBSF (125, 250, 500 μM) for 4 h, or pre-treated with the same increasing doses of AEBSF in (A) for 4 h, and then co-treated with 4 h of 20 μM CCCP or 10 μM oligomycin. Whole-cell lysates were analyzed by WB. **c** Mitochondrial fractionation of WT HEK293T cells lysates following 4 h of DMSO or CCCP (20 μM) and analyzed by WB. **d** SC, shHtrA2 A, B and C HEK293T cells were treated with DMSO, CCCP (20 μM) or oligomycin (10 μM) for 4 h. Whole-cell lysates were analyzed by WB. **e** SC and shHtrA2 C HEK293T cells were transfected with 0.4 μg of empty plasmid, WT HtrA2 plasmid or S306A HtrA2 plasmid overnight, and then treated with 4 h of DMSO or oligomycin (10 μM). Whole cell lysates were analyzed by WB. **f** SC and shHtrA2 B HEK293T cells were transfected with 0.4 μg of empty plasmid, WT HtrA2 plasmid, S141A HtrA2 plasmid or G399S HtrA2 plasmid overnight, and then treated with 4 h of DMSO or oligomycin (10 μM). Whole cell lysates were analyzed by WB.

mitochondrial fractions to study the subcellular location of HtrA2 and OMA1 following MPIS (Fig. 6c). The mature HtrA2 band was found in both fractions, with higher intensity in the mitochondrial fraction. Interestingly, the ~55 kDa pre-sequence of HtrA2 was retained in the cytosolic fraction following CCCP treatment, which indicated on-going MPIS. Such cytosolic retention of mitochondrial pro-protein was also reported in our previous study, where cytosolic pro-NLRX1 was found to induce mitophagy following MPIS[6]. Unlike HtrA2, OMA1 was found almost exclusively in the mitochondrial fraction, and following CCCP, the mature form of OMA1 was again degraded while pro-OMA1 was not noticeably retained in the cytosol. Similar to CCCP, oligomycin treatment also led to cytosolic retention of pro-HtrA2, although the effect seemed to be weaker (Supplementary Fig. 5a). After we confirmed that HtrA2 was localized in both cytosol and mitochondria following MPIS, we used three HtrA2-targeting shRNAs to study the effect of HtrA2 silencing on the cleavage of DELE1 after MPIS (Fig. 6d). Strikingly, CCCP and oligomycin-induced ATF4 and DELE1 cleavage were found to be dampened in HtrA2-silenced cells, with the effect more obvious on DELE1-VS than DELE1-S. The reduced ISR activation was also captured in HtrA2 knockout MEF cells that were treated with CCCP or oligomycin, as the level of ATF4 and CHOP were both reduced when HtrA2 was removed (Supplementary Fig. 5b, c). Moreover, the OMA1-directed cleavage of OPA1 was not affected by HtrA2 knockdown (Supplementary Fig. 5d), suggesting that OMA1 was functioning normally in HtrA2 knockdown cells and that the two proteases were not relying on each other. Consistent with our data in HEK293 cells, shHtrA2 constructs B and C led to efficient HtrA2 knockdown in HeLa cells, and the resulting inhibition on DELE1 cleavage was more robust on DELE1-VS as compared to the DELE1-S form (Supplementary Fig. 5e). As was the case for OMA1 (see Supplementary Fig. 4c), HtrA2 protein levels were comparable between HEK293T and HeLa cells (Supplementary Fig. 5f). To further study the role of HtrA2 in the generation of DELE1-VS, we created a HtrA2 mutant where the catalytically active serine 306 residue was substituted with an alanine (S306A HtrA2)[39]. When we over-expressed WT HtrA2 and the catalytically dead S306A HtrA2 in shHtrA2 cells, following a challenge of oligomycin, the dampened DELE1-VS could be rescued by WT HtrA2 but not S306A HtrA2 (Fig. 6e). Together, these results suggest that HtrA2 is a novel DELE1 protease and that this protease may preferentially mediate DELE1 cleavage at the "VS" site during non-depolarizing MPIS, while playing a relatively minor role in the cleavage that forms DELE1-S.

Lastly, given the known link between HtrA2 and the development of PD, we generated the PD-related S141A and G399S HtrA2 mutants[29,30] and investigated the potential effect of these two mutations on the cleavage of DELE1 in HEK293T cells (Fig. 6f). Interestingly, in shHtrA2 B cells, while over-expression of WT and S141A HtrA2 was able to partially rescue the levels of DELE1-VS after oligomycin treatment, this rescue was blunted in cells expressing the HtrA2 G399S variant, suggesting that this PD-associated mutation might hamper ISR induction upon mitochondrial stress. Taken together, these results suggest that, in both HEK293T and HeLa cells, HtrA2 plays a critical role in DELE1 cleavage and ISR activation following MPIS.

## Discussion

In this work, we have demonstrated the important roles of DELE1 and HRI in the relay of MPIS to the ISR at the endogenous level. We report here for the first time the cleavage of endogenous DELE1 in response to MPIS, and identify the formation of a previously uncharacterized DELE1-VS form induced by non-depolarizing MPIS. In addition to chemical inducers such as CCCP, oligomycin and MB-6/MB-10, we also triggered MPIS by overloading the mitochondrial protein import machinery through either over-expressing mitochondrial proteins or silencing key import machinery by RNAi. We identified that the knockdown of TIM23, which is required for importing mitochondrial proteins to the matrix, triggered the most robust DELE1 cleavage and ATF4 induction. We also provide evidence that the induction of the ISR and mitophagy pathways are not inter-dependent during MPIS. Our data further suggest that the DELE1 cleavages at the sites "S" and "VS" occur in the cytosol and while the protein is stuck in the entry

machinery, respectively. Finally, our data suggest that OMA1 could be dispensable for DELE1-S and -VS cleavage in certain cell lines, and demonstrate a critical role for the mitochondrial protease HtrA2 in the formation of the DELE1-VS form.

The discovery of the DELE1-VS form of DELE1, generated only during non-depolarizing MPIS, was unexpected as previous work that identified the cleavage of DELE1 into DELE1-S in response to mitochondrial stress failed to identify DELE1-VS[10–12]. This is likely explained by the fact that previous investigations did not detect the endogenous DELE1 through a dedicated antibody like we did in this study, but instead relied on the detection of a C-terminal HA tag in cells expressing DELE1-L-HA[10–12]. Since the "VS" cleavage site is very close to the C-terminal end of the protein, the formation of the DELE1-VS form would not have been possible using an anti-HA antibody. This illustrates the importance of studying DELE1 processing during MPIS at the endogenous level. Furthermore, the discovery of the "VS" cleavage site allowed to refine the model of DELE1 cleavage during depolarizing vs non-depolarizing MPIS by suggesting that the "VS" cleavage occurs at the mitochondria, most likely when the protein is stuck in the entry machinery, while the "S" cleavage appears to occur in the cytosol (see model in Fig. 5g). Regarding the site of cleavage that generated DELE1-VS, our data suggest that the most probable site lies between amino acids 456-466, but we failed to identify an individual amino acid substitution that would completely abolish the formation of DELE1-VS. The seemingly promiscuous nature of DELE1-VS cleavage reminded us of the cleavage that generates DELE1-S, which was identified to be histidine 142[10], but the substitution of this residue did not fully abolish the formation of DELE1-S. Overall, these results suggest that the protease(s) that are responsible for cleaving DELE1 likely do not have a preference for a specific residue. Nevertheless, we have showed that DELE1-VS is capable of interacting with HRI, which suggests that it may has a physiological function in the activation of ISR. Compared to DELE1-S, DELE1-VS would lack the last TPR repeat which is located at AA 472-504[10,12]. A previous report[10] has created a DELE1 mutant that lacks the last TPR repeat by removing the last 46 amino acids (Δ469-515) and demonstrated that the removal of the last TPR repeat did not abolish the ability of such DELE1 mutant to activate ATF4 reporter. It is therefore likely that DELE1-VS would function similarly to DELE-S in the activation of ISR, but whether DELE1-VS maintains the ability to form oligomers[40] is currently unknown.

Our recent studies have demonstrated that MPIS is an overarching stress that triggers induction of mitophagy[6]. During MPIS, the mitochondrial protein NLRX1, which normally translocates to the mitochondrial matrix, was retained into the cytosol, and this retention served as a danger signal to trigger mitophagy[6]. Once in the cytosol, we found that NLRX1 engages in a complex with the endoplasmic reticulum (ER) protein RRBP1, which is also recruited (independently from NLRX1) to the mitochondria-ER interface during MPIS, and that the NLRX1-RRBP1 complex further recruits the protein LC3 to initiate the first steps of mitophagy targeting[6]. Because MPIS appears to serve as a common trigger for both mitophagy and the mitochondrial ISR, we aimed to determine if these two processes were inter-dependent. Our results suggest that the ISR and mitophagy might represent different cellular responses to early and late stages of MPIS. The results from our study indicated that the ISR usually peaked at 4 h after the induction of MPIS, whereas our previous work showed that mitophagy was most strongly triggered after prolonged treatment[6]. We speculate that the ISR might be utilized by the cell as an early response to recover from MPIS, since temporally inhibiting cytosolic protein synthesis would help to reduce the workload on the mitochondrial protein import machinery and prevent the cytosolic buildup of more mitochondrial presequence-containing proteins. In contrast, if MPIS persists and the stress to the mitochondria is deemed beyond repair, mitophagy would be induced to remove the dysfunctional mitochondria and prevent more damage to the rest of mitochondrial network and to the cell. Moreover, it is interesting to note that the DELE1-HRI pathway mediates cellular responses to MPIS induced by inhibition of the TIM23 complex while the NLRX1-RRBP1 axis mediates mitophagy responses to MPIS that can be recapitulated by silencing of

*TOM20* but not the *TIM23*[6]. This suggests that, in addition to kinetics differences as discussed above, the DELE1-HRI and NLRX1-RRBP1 pathways may mediate responses to different types of MPIS, depending on the step at which mitochondrial protein import is inhibited.

The fact that OMA1 appeared to be dispensable for DELE1 cleavage during depolarizing or non-depolarizing MPIS in HEK293T cells was surprising, given that this mitochondrial protease was shown to play a key role in DELE1 processing in previous studies[10–12]. While we do not dispute the fact that OMA1 cleaves DELE1 during MPIS and could even be the most critical DELE1 protease in some cellular systems as we observed here in HeLa cells, we believe that the key role played by HtrA2 (and possibly other proteases) in our cellular model of HEK293T cells may have masked any contribution of OMA1, as OMA1 would then appear to be dispensable, as we reported in our assays. Our data instead suggest that several proteases, such as OMA1 and HtrA2 and possibly others, may be able to cleave DELE1 during mitochondrial stress. In this scenario, it would be more the location of DELE1 (normal translocation versus cytosolic retention), rather than engagement of a specific protease, which would be the limiting factor that dictates DELE1 processing and activation of the ISR. Upon improper delivery of DELE1 to the mitochondrial matrix, partially unfolded regions of DELE1 such as those we identified in the Alphafold model (see Fig. 3f) could become permissive towards protease-mediated cleavage without an absolute requirement on the nature of the protease, as long as the protease is also active during MPIS. In line with this, our data showed that, during depolarizing MPIS, the proform of HtrA2 was retained into the cytosol, suggesting that the co-retention of both DELE1 and HtrA2 may allow formation of the DELE1-S form, which serves as a signal to induce the ISR. During non-depolarizing MPIS, the cytosolic retention of HtrA2 appeared much weaker, which could favor an initial cleavage of DELE1 on the "VS" site in the IMS, where mature HtrA2 (and OMA1) resides, followed by a cleavage on the "S" site once the mistargeted DELE1 is sent back to the cytosol. Finally, it is interesting to note that both OMA1 and HtrA2 are mitochondrial proteases that have been associated with mitochondrial stress responses and future work will identify if other proteases, associated or not with mitochondrial stress, can also cleave DELE1 in the cytosol upon retention caused by MPIS.

The identification of HtrA2 as a DELE1 protease active during MPIS can be potentially important in the context of PD. Indeed, mutations in *HTRA2*, including G399S, A141S and P143A have been linked to familial forms of PD[33–35]. Additionally, *mnd2* mice, which were identified three decades ago[31,32] and studied as a model of neurodegeneration in rodents, develop the disease as a result of a mutation in the *Htra2* gene. Up until now, the research on HtrA2 and PD essentially focused on the link between HtrA2 and apoptosis during mitochondrial stress, since it was reported that, upon cytosolic retention, HtrA2 can act as a pro-apoptotic signal; indeed, cytosolic HtrA2 can cleave Inhibitor of Apoptosis Proteins (IAPs), such as c-IAP1 and XIAP, thereby promoting apoptosis[37,38]. Our findings now suggest that MPIS and the regulation of the ISR may be differentially regulated in patients carrying mutations in *HTRA2*, and in particular, the G399S variant of HtrA2 may affect DELE1 cleavage into the DELE1-VS form upon non-depolarizing MPIS. Future studies will thus aim at identifying if mutations in *HTRA2* affect DELE1 cleavage in neurons or other cell populations of the central nervous system in conditions that induce MPIS. In support of this notion, our previous studies[8] have identified a role for HRI, the ISR kinase acting immediately downstream of DELE1, in controlling the aggregation of alpha-synuclein, an event associated with PD pathogenesis. The generation and characterization of *Dele1* deficient mice will likely represent a critical step towards the elucidation of the role played by the DELE1-HRI axis in vivo and in murine models of neurodegeneration.

## Materials and methods
### Cell lines
HEK293T, HCT116, HeLa and 3T3 MEF cells were maintained at 37 °C with 5% CO$_2$ in Dulbecco's modified Eagle's medium (DMEM) with 10% heat-inactivated fetal bovine serum (FBS) and 1% penicillin/streptomycin. DMEM (319-005-CL), FBS (098-150), 0.25% Trypsin-EDTA (325-043-EL) and penicillin/streptomycin (450-201-EL) were purchased from Wisent INC. All cell lines were split and seeded in new culture flask/dish with a 1/5 ratio once they reached around 90-100% confluency. Unless specifically indicated, all treatments were done within 24 h post seeding in fresh culture media. All cell lines were routinely checked for Mycoplasma contamination by PCR.

### Chemicals
CCCP (C2759, Sigma), Puromycin (P8833, Sigma), MitoBloCK-6 (5.05759.0001, EMD Millipore), MitoBloCK-10 (HY-115467, MedChemExpress), FuGENE HD Transfection Reagent (E2311, Promega), Oligomycin A (75351, Sigma), Antimycin A (A8674, Sigma), MG132 (1748, Tocris), *o*-phenanthroline (516705, Sigma), AEBSF (A8456, Sigma), TMRE (T669, Thermo Fisher Scientific) and trans-Resveratrol (70675-50, Cederlane).

### Antibodies
ATF4 (11815S, Cell Signaling, 1/2000), DELE1(PA5-57712, Thermo Fisher Scientific, 1/2000), HRI (20499-1-AP, Proteintech, 1/3000), GAPDH (5174S, Cell Signaling, 1/10,000), HtrA2 (#9745, Cell Signaling, 1/1000), OMA1 (HPA055120, Sigma, 1/2500), OPA1 (612606, BD, 1/2500), NLRX1 (MA5-27207, Thermo Fisher Scientific, 1/3000), RRBP1 (PA5-21392, Thermo Fisher Scientific, 1/2000), Tubulin (T9026, Millipore Sigma, 1/10,000), VDAC1 (ab14734, abcam, 1/2000), COX4 (ab33985, abcam, 1/4000), LC3B 3868S, Cell Signaling, 1/1000), PINK1 (BC100-494, Novus, 1/1000), TOMM20 (42406, Cell Signaling, 1/2000), TOM40 (18409-1-AP, Proteintech, 1/10,000), TOM70 (14528-1-AP, Proteintech, 1/10,000), TIMM22 (14927-1-AP, Proteintech, 1/2500), MIA40 (21090-1-AP, Proteintech, 1/2500), TIMM23 (11123-1-AP, Proteintech, 1:2000), HA Tag (ab18181, abcam, 1/1000), FLAG Tag (14793S, Cell Signaling, 1/1000), LONP1 (15440-1-AP, Proteintech, 1/1000), Myc-tag (2276S, Cell Signaling, 1/1000), HRP Rabbit (111-035-003, Jackson ImmunoResearch Laboratories, 1/10,000), HRP mouse (115-035-003, Jackson ImmunoResearch Laboratories, 1/10,000), Alexa fluor 488 rabbit (A11034, Life Technologies, 1/300), Cy3-conjugated goat anti-mouse (115-165-003, Jackson ImmunoResearch Laboratories, 1/300).

Note: We have been informed by Thermo Fisher Scientific, the supplier of the anti-DELE1 antibody that we used throughout the manuscript that this antibody was not available anymore, as the catalog number (PA5-57712) has been discontinued.

### Cell lysate preparation for whole cell lysis and hot SDS collection
For normal whole-cell lysis, HEK293T cells were collected in phosphate-buffered saline (PBS). Cells were centrifuged at $10,000 \times g$ followed by removing PBS. Cell pellets were lysed in SDS-RIPA buffer (supplemented with Protease Inhibitor Cocktail P8340, Sigma, and Phosphatase Inhibitor Cocktail 78420, Thermo Fisher Scientific), centrifuged at $10,000 \times g$ to remove insoluble fraction. For hot SDS collection, HEK293T cells were removed from cell culture plate after adding boiling hot SDS laemmli buffer (6x western loading buffer), and subsequently boiled at 95 °C for 15 min.

### Mitochondrial fractionation
HEK293T cells were collected in ice-cold phosphate-buffered saline (PBS). Cells were centrifuged at $10,000 \times g$ followed by removing PBS. Cells were lysed using a 22 G needle and syringe in mitochondrial isolation buffer MIB: (210 mM mannitol, 70 mM sucrose, 1 mM EDTA, and 10 mM Hepes (pH 7.5), supplemented with Protease Inhibitor Cocktail P8340, Sigma and Phosphatase Inhibitor Cocktail 78420, Thermo Fisher Scientific). Lysates were centrifuged at $700 \times g$ to remove the plasma membrane, and then centrifuged at $6000 \times g$ to isolate mitochondria/heavy membrane fraction. Isolated mitochondrial/heavy membrane pellets were subsequently lysed in mitochondrial lysis buffer (MLB: 50 mM Tris-HCl (pH 7.5), 100 mM NaCl, 10 mM MgCl$_2$, 0.5% NP40, 10% glycerol, and supplemented with Protease

Inhibitor Cocktail P8340, Sigma and Phosphatase Inhibitor Cocktail 78420, Thermo Fisher Scientific).

## Western blots

For Western blot detection of protein targets such as DELE1, cells were collected in ice-cold PBS and lysed in SDS-RIPA buffer, mitochondria isolation buffer or mitochondrial lysis buffer supplemented with Protease and Phosphatase Inhibitor Cocktail. Cell lysates were then added with Laemmli blue loading buffer and boiled at 95 °C for 5 min. Lysates were then run on 7% or 12% acrylamide gels (BioRad, 1610148) and transferred onto FluoroTrans® W PVDF membranes (VWR, CA29301-856) by Trans-Blot Turbo Transfer System from BioRad, with a setting of 15 V and 1.5 A for 60 min. PVDF membranes were first incubated with 5% skim milk (Bio-Shop Canada, SKI400) dissolved in Tris-Buffered Saline with 0.1% Tween® 20 Detergent (TBST) as blocking solution for 1 h and then incubated with the corresponding antibodies (see above for suppliers, catalog numbers and concentrations) overnight at 4°. All antibodies were diluted with 5% Bovine Serum Albumin (BSA, purchased from Sigma, catalog number A7030) dissolved in TBST. After overnight incubation, membranes were washed three times with TBST (15 min each) and incubated with 1:10,000 anti-mouse or anti-rabbit HRP-conjugated antibodies in room temperature for 1 h. After three more 15-min washes with TBST, membranes were incubated with Classico or Crescendo Luminata Western HRP substrate (Thermo Fisher Scientific, WBLUC0500 and WBLUR0500) for 3 to 5 min and images were taken by a ChemiDoc imaging system.

## Immunofluorescence

HeLa cells were fixed with methanol for 10 min. Sample preparation of immunofluorescence assays was performed as described[41]. Images were taken on a Zeiss LSM 700 confocal microscope with Zen™ software.

## Co-Immunoprecipitation

Protein G beads (Pierce 88805, Thermo Fisher Scientific) were washed twice with mitochondria lysis buffer and centrifuged at 300 g for 5 min before adding cell lysate and antibodies. The immunoprecipitation reaction was incubated for 2 h at 4 °C. Next, samples were centrifuged at 10,000 g for 5 min, washed and resuspended in mitochondria lysis buffer with laemmli buffer and boiled for 5 min.

## Virus production and transfections

For lentiviral vector production, HEK293T cells were transfected with psPAX2 (1.75 µg) and pMD2.G (0.875 µg) in combination with the shRNA pLKO.1 plasmid of interest (2.625 µg). These plasmids were added to 300 µl of basal DMEM, mixed and then added with FuGene HD (4:1 ratio). After 15 min the transfection mix were added to the cells. After 48 h, the culture fluid was collected, spun down at 650 g to remove cell debris and stored at −80 °C. HEK293T cells were then transduced with the collected supernatant and were maintained under puromycin selection (3 µg/mL).

The shRNA sequences used in this study are: shRRBP1 CCTAATGGGAAGATACCTGAT, shNLRX1 GAGGAGGACTACTA-CAACGAT, shPINK1 CGGCTGGAGGAGTATCTGATA, shHRI CAA-GAGGCTGTCAAGTCGTC (construct A), AGCTACTTTGCCAGACG TTTA (construct B), shDELE1 ACAGCAAAGCGCAGTACAATG (construct A) TGAGGAAGGTCCCGGTGATTT (construct B), shOMA1 GAAGTGCTTTGTCATCTAATT (construct A), ATGGAAGCTATT CCTTGGTTTT (construct B), shHtrA2 CTGATCGTCTTCGAGAGTTTC (construct A), GCTGAACTACAGCTTCGAGAA (construct B), CATG GTGTACTCATCCATAAA (construct C).

## siRNA knockdown

HEK293T cells were seeded in antibiotic-free DMEM with Lipofectamine™ RNAiMAX Transfection Reagent and siRNA. Cells were collected 2- or 4-day post-transfection. The siRNA sequences used in this study are obtained from Thermo Fisher Scientific, and the catalog numbers are: Negative control AM4620, siTOMM20 4392420-s18950,

siTOMM40 4392420-s20450, siTOMM70 4392420-s19108, siMIA40 4392420-s43607, siTIMM22 4392420-s26724 and siTIMM23 4390824-s223113.

## Statistics and reproducibility

Experiments are independently repeated and the results are reproducible. The data shown in the article are representative of the number of individual experiments listed:

Figure 1a, $n = 5$; Fig. 1b, $n = 2$; Fig. 1c, $n = 3$; Fig. 1d, $n = 3$; Fig. 1e, $n = 3$; Fig. 1f, $n = 3$; Fig. 1g, $n = 3$; Fig. 1h, $n = 2$; Fig. 2a, $n = 2$; Fig. 2b, $n = 2$; Fig. 2c, $n = 2$; Fig. 3a, $n = 2$; Fig. 3c, $n = 3$; Fig. 3d, $n = 3$; Fig. 3e, $n = 3$; Fig. 3h, $n = 2$; Fig. 4a, $n = 3$; Fig. 4b, $n = 3$; Fig. 4c, $n = 3$; Fig. 4d, $n = 3$; Fig. 4e, $n = 3$; Fig. 4f, $n = 3$; Fig. 5a, $n = 3$; Fig. 5b, $n = 3$; Fig. 5c, $n = 3$; Fig. 5d, $n = 3$; Fig. 5e, $n = 3$; Fig. 5f, $n = 3$; Fig. 6a, $n = 2$; Fig. 6b, $n = 3$; Fig. 6c, $n = 3$; Fig. 6d, $n = 3$; Fig. 6e, $n = 2$; Fig. 6f, $n = 2$. Supplementary Fig. 1a, $n = 2$; Supplementary Fig. 1a, $n = 2$; Supplementary Fig. 1b, $n = 2$; Supplementary Fig. 1c, $n = 3$; Supplementary Fig. 1d, $n = 2$; Supplementary Fig. 1e, $n = 2$; Supplementary Fig. 1f, $n = 2$; Supplementary Fig. 1g, $n = 2$; Supplementary Fig. 1h, $n = 2$; Supplementary Fig. 2, $n = 2$; Supplementary Fig. 3a, $n = 2$; Supplementary Fig. 3b, $n = 2$; Supplementary Fig. 3c, $n = 3$; Supplementary Fig. 3d, $n = 2$; Supplementary Fig. 3e, $n = 3$; Supplementary Fig. 3f, $n = 2$; Supplementary Fig. 3g, $n = 3$; Supplementary Fig. 4a, $n = 2$; Supplementary Fig. 4b, $n = 2$; Supplementary Fig. 4c, $n = 2$; Supplementary Fig. 4d, $n = 2$; Supplementary Fig. 4e, $n = 2$; Supplementary Fig. 5a, $n = 2$; Supplementary Fig. 5b, $n = 4$; Supplementary Fig. 5c, $n = 4$; Supplementary Fig. 5d, $n = 3$, Supplementary Fig. 5e, $n = 2$, Supplementary Fig. 5f, $n = 2$.

## Reporting summary

Further information on research design is available in the Nature Portfolio Reporting Summary linked to this article.

## Data availability

This study does not generate publicly available datasets. The data that supports the findings of this study are available in the article, figures and the Supplementary Figs. Uncropped and unedited blot images are provided in Supplementary Fig. 6.

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

## Acknowledgements
This study was supported by Canadian Institutes of Health Research (CIHR) and Agence Nationale de la Recherche (ANR, MITOCISR).

## Author contributions
S.E.G., P.Y.B. and S.A.K. designed the experiments. S.A.K. performed some of the HCT116, HeLa and IF experiments. D.A. performed some of the over-expression and MEF experiments. L.S. performed some of the shRNA knockdown experiments. P.Y.B. performed all other experiments. P.Y.B., D.J.P. and S.E.G. designed the manuscript's structure and organization. P.Y.B. wrote the manuscript, which was edited by S.A.K., D.J.P. and S.E.G.

## Competing interests
The authors declare no competing interests.
