## [Peer review file · Communications Biology]

Reviewers' comments:

Reviewer #1 (Remarks to the Author):

Comments to the authors

In 2020, two groups independently reported that DELE1 activates HRI-mediated integrated stress response (ISR) upon mitochondrial stress. In their model, DELE1 is cleaved by OMA1 upon mitochondrial stress, and cleaved DELE1 (DELE1-S) is released into the cytosol where it interacts with and activates cytosolic kinase HRI through its C-terminal TPR domains. In this manuscript, the authors found additional cleavage of DELE1 by HtrA2 presumably at DELE1 C-terminus (DELE1-VS). This newly identified DELE1 cleavage was induced specifically under mitochondrial import stress (MPIS), but not under mitochondrial stress that is accompanied with mitochondrial depolarization. The previous DELE1 detection system relied on its C-terminal HA tag, and thus it overlooked the C-terminal cleavage that was newly identified in this manuscript. Therefore, the main finding of this study (the HtrA2-mediated generation of DELE1-VS upon MPIS) is very interesting. However, it is not investigated whether the DELE1-VS generation plays critical roles in the MPIS-induced ISR activation. Also, some claims that show discrepancy with previous two Nature papers need more careful investigation, as these are relatively big claims. Particularly, regarding the possibility that DELE1-S is generated in the cytosol, more careful experimental design and data interpretation should be considered.

Major points

1. In Fig. 1, the authors mentioned the rapid turn-over of endogenous DELE1 protein. As this observation was already reported in the recent study (Fessler et al., Nature Commun., 2022), they should cite this paper in the body of manuscript.
2. In Fig. S2C, the authors claimed that DELE1 and HRI are not involved in the mitochondrial stress-induced ISR activation in HeLa cells. Although it looks their shRNA system efficiently suppressed both of HRI and DELE1 expression, I strongly recommend the authors to confirm their finding by siRNAs or even by creating CRISPR/Cas9-mediated knockout clones. Although the authors stated that HRI shRNA did not affect the endogenous DELE1 level in HEK293T cells (Fig. 1C), it seems HRI shRNA significantly reduced the endogenous DELE1 in HeLa cells (Fig. S2C), which is confusing, and raises several concerns, for example, several off-targets effects by these shRNAs. Any explanations for this?
3. In Fig. 1F, MG132 treatment clearly induces DELE1-VS, which is interesting. Although the authors interpreted this data as follows; "Thus, insufficient clearance of misfolded proteins by the proteasome system likely results in protein clogging at the site of protein import into mitochondria, thereby triggering the HRI-DELE1 axis of the ISR." However, we cannot rule out the possibility that DELE1-VS itself is rapidly degraded by proteasome. The authors should experimentally dissect these two possibilities.
4. Related to Fig. 1F, MG132-induced ISR activation is actually abrogated by the deletion of DELE1 and HRI?
5. In Fig. 3, the authors tried to estimate the actual cleavage site that generated DELE1-VS. However, their conclusion (DELE1-VS generated by both cleavage at N-terminus and C-terminus) is not fully supported by their data. As this is a very critical point, the actual cleavage site should be determined by other methods, for example, mass spec analysis.
6. In Fig. 4, the authors stated that MPIS-induced ISR and mitophagy are independently induced. I

agree with the importance of this observation, but it blocks the entire flow of the manuscript. Therefore, I recommend the authors to move these data to supplemental figures, and keep to a minimum description in the body of the manuscript.

7. In Fig. 5, the authors stated that OMA1 is not involved in DELE1 cleavage in HEK293T cells, but mediates the cleavage in HeLa cells. The authors only show the DELE1 cleaved form (S and VS) on their immunoblot in main Fig. 5 that addressed the OMA1 dependency in HEK293T cells, but they should also show the behavior of DELE1 full-length form. In Fig. S4A and S4B that addressed OMA1 dependency in HeLa cells, the authors showed the DELE1 full-length form. Although both DELE1-S and -VS forms reduced after OMA1 shRNA treatment, the recovery of the DELE1 full-length form was not observed under this condition, which is really confusing. Where is the un-cleaved DELE1 full-length form? This point should be carefully re-evaluated. Otherwise, the current experimental setting and data interpretation are too premature to conclude the cell-type-dependent OMA1 dependency for DELE1 cleavage.

8. As mentioned in comment 2, the OMA1 dependency for DELE1 cleavage should be also confirmed by other gene suppression methods like siRNAs or CRISPR/Cas9-mediated gene editing.

9. In the body of manuscript on page 9, about the disappearance of OMA1 during CCCP treatment, the authors only mention YME1L involvement in OMA1 degradation. In addition, it is known that OMA1 degrades through self-cleavage mechanism. Therefore, following two reports also should be cited;

Baker, M.J., Lampe, P.A., Stojanovski, D., Korwitz, A., Anand, R., Tatsuta, T., and Langer, T. (2014). Stress-induced OMA1 activation and autocatalytic turnover regulate OMA1-dependent mitochondrial dynamics. *EMBO J* 33, 578-593. [10.1002/embj.201386474](https://doi.org/10.1002/embj.201386474).

Zhang, K., Li, H., and Song, Z. (2014). Membrane depolarization activates the mitochondrial protease OMA1 by stimulating self-cleavage. *EMBO Rep* 15, 576-585. [10.1002/embr.201338240](https://doi.org/10.1002/embr.201338240).

10. In Fig. 6B, the authors argued that unlike O+A or CCCP, DELE1-S and -VS forms retained mitochondria fraction under Oligomycin-treated condition. However, the total amount of DELE1 is not equivalent among these stimuli in this figure. It looks like a larger amount of DELE1-S and -VS forms are generated under Oligomycin treatment compared to O+A and CCCP stimuli. Thus, it is hard to interpret data as the authors did. In other figures, for example, in Fig 1B, DELE1 band shows similar intensity between CCCP and Oligomycin treatment on immunoblot. The authors repeat this experiment to confirm their conclusion.

11. In the body of manuscript on page 10 and Fig. 6F, the authors mentioned the possibility that DELE1-S form is not generated by OMA1 cleavage but it may be generated in the cytosol through unknown mechanisms. However, the current data sets and their data interpretation do not support their conclusion. First, their discussion (line 6-9, on page 10) is too speculative. Although DELE1-S form appears rapidly in the cytosol (Fig. 6A), the authors' interpretation of this data lacks scientific logic as no one tested the actual speed of retrotranslocation of mitochondrial proteins so far. Second, to support their conclusion experimentally, they examined whether DELE1 mutant that lacks "MTS" can be cleaved or not in Fig. 6C, but one important point should be clarified regarding experimental setting in this figure. Which region is deleted as "MTS" here? The authors should indicate the precise information of this mutant in the manuscript. It has been reported that DELE1 "MTS" has a unique feature – N-terminal 30 a.a. deletion does not prevent the mitochondrial localization of DELE1, and at least N-terminal 101 a.a. deletion is required for the efficient prevention of its mitochondrial localization (Fessler et al., *Nature*, 2020 and Guo et al., *Nature*, 2020). If the authors only deleted N-terminal small portion of DELE1 in this experiment, they should use DELE1 (d101) mutant.

12. As pointed out in comment 7, the authors should also examine whether the DELE1 full-length is recovered after AEBSF treatment in Fig. 7A and 7B. Otherwise, we cannot rule out the possibility that total expression of DELE1 is reduced by AEBSF for unknown reasons. This is also the case for HtrA2 knockdown in Fig. 7D and 7E. Of note, the recovery of the DELE1 full-length form was not observed under HtrA2 depleted condition in Fig. S5C, again which is confusing.

13. In Fig. 7E, the authors also need to test the protease-dead mutant of HtrA2 in addition to reported PD-related HtrA2 mutants.

14. Again, the generation of DELE1-VS form under MPIS is really exciting finding. However, currently, the actual roles of this DELE1-VS form are totally unknown. The knockdown of HtrA2 attenuates the MPIS-induced ISR activation? Does DELE1-VS interact with HRI as DELE1-S does? The actual cleavage site in DELE1 C-terminus is not determined, but if the cleavage happens somewhere within C-terminal TPR domains, that cleavage may interrupt the proper folding TPR domains and thus it can disable the HRI binding. In that case, DELE1-VS form generation may have other meanings rather than ISR activation.

Reviewer #3 (Remarks to the Author):

In their manuscript titled "Cytosolic retention of the mitochondrial protease HtrA2 during mitochondrial protein import stress (MPIS) triggers the DELE1-HRI pathway," Bi et al. present that mitochondrial protein import stress is an overarching stress that triggers the DELE1-HRI-mediated integrated stress response. Using an antibody against DELE1, they detected distinct cleavage patterns of endogenous DELE1 depending on the types of MPIS. While only one short form of DELE1 is generated under depolarizing MPIS, under non-depolarizing MPIS such as the oligomycin condition, the authors uncovered an additional, even shorter form of DELE1 (DELE1-VS) generated by cleaving at the C-terminus of DELE1 by HtrA2 while stuck in the translocase pore upon import. While the study shows some important biology of endogenous DELE1, a critical protein that relays mitochondrial dysfunction to the ISR, further evidence is required to solidify the molecular mechanisms of DELE1-mediated ISR activation proposed in this paper.

More detailed comments:

1. This paper relies on an anti-DELE1 antibody (PA5-57712) to detect DELE1. However, according to the product information on the website (<https://www.thermofisher.com/antibody/product/DELE-Antibody-Polyclonal/PA5-57712>), this antibody gives rise to a strong signal in the nucleus in IHC experiments, suggesting strong non-specificity of this antibody. Although the antibody's specificity has been partially validated by shRNA knocking down of DELE1, it would be critical to further validate its specificity using a knockout (KO) cell line.
2. Using shRNA knockdown (KD) of important translocase components such as TOMM40, TOMM20, TOMM70, TIMM23, TIMM22, and MIA40 to trigger the MPIS, only knockdown of TIMM23 induces DELE1 cleavage and ATF4 activation. This contradicts previously reported data in Figure 6d and e (<https://doi.org/10.1038/s41467-022-29479-y>). TOMM40 is known as the main translocase pore, and its knockdown would block mitochondrial protein import. The data regarding MIA40 knockdown is not consistent with the MB-6 results. Therefore, the authors should provide an explanation and reconcile these observations.

3. In Figure 5, the authors conclude that OMA1 is not required for DELE1 cleavage in HEK293 cells based on the results of shRNA knockdown of OMA1. Although the OMA1 knockdown cells showed a significant reduction in OMA1 protein levels, they still exhibited OPA1 cleavage with slower kinetics, as shown in Figure 5c. This suggests the presence of residual OMA1 activity, which might be sufficient to cleave DELE1. To achieve a clearer and more definitive interpretation, the authors should employ an OMA1 knockout (KO) cell line. This would provide better control and eliminate any potential residual activity of OMA1, leading to more robust conclusions.

4. In Figure 6, the authors interpret the accumulation of DELE1-s in the cytosolic fraction upon CCCP treatment as a result of DELE1 cleavage in the cytosol. However, it is insufficient to conclude this, as it is possible that cleavage and release of DELE1 from mitochondria can occur simultaneously under CCCP treatment, while at a slower release rate under oligomycin treatment. Therefore, alternative explanations should be considered for the observed accumulation of DELE1-s in the cytosolic fraction upon CCCP treatment, and further experiments or analyses are needed to solidify the proposed mechanism.

5. It is indeed very interesting that the DELE1-VS form is generated upon non-depolarizing MPIS. However, the physiological role of this fragment remains less clear. According to the model proposed in Figure 6, the cleavage of DELE1 at the C-terminus may be required for its release to the cytosol, where it is further cleaved at the "s" site to generate the very short form. However, this model does not explain why, under oligomycin treatment conditions, the majority of DELE1 exists as the short form. While Figure 7 demonstrates the role of HtrA2 in generating DELE1-VS, it would be important to investigate whether reducing the very short form of DELE1 abolishes the induction of ATF4 activation through ISR. Such experiments would provide valuable insights into the functional significance of the different DELE1 forms and their contributions to the integrated stress response.

Point-by-point response to reviewers' comments and summary of new data:

We would like to thank the reviewers and the editor for the time and effort spent on going through and giving insightful comments on how we could improve the manuscript. Please find below:

- a. The summary of the new data that we have included in the revised manuscript along with changes we have made.
- b. The point-by-point response to the reviewers' comments.

a) Summary of new data:

- Demonstrated the potential range of DELE1-VS cleavage site is most likely within AA 456-466 by generating DELE1 mutants with serial deletions in the c-terminus region and monitoring the formation of DELE1-VS following non-depolarizing MPIS. (Fig. 3e)
- Included co-immunoprecipitation assay to demonstrate the interaction of DELE1-S and -VS with HRI. (Fig. 3h)
- Investigated the cleavage of DELE1-VS by WT HtrA2 and the catalytically-dead HtrA2 S306A mutant, to further confirm the role played by HtrA2 in the generation of DELE1-VS. (Fig. 6e)
- Included blots of ATF4 in HtrA2 knockdown cells and HtrA2 knockout 3T3 MEF cells treated with MPIS inducers to demonstrate the importance of HtrA2 in ISR activation following MPIS (Fig. 6d, Supplementary Fig. 5b-c)
- Included a recovery assay of CCCP and oligomycin with or without MG-132 to study the half-life of various DELE1 species, and investigated the potential involvement of proteasome in the degradation of DELE1. (Supplementary Fig. 1f-g)
- Included shRNA silencing of LONP1 to illustrate the importance of LONP1 during the degradation of full-length DELE1 that are imported to the matrix of mitochondria under baseline conditions. (Fig. 5a)
- Included additional shRNA constructs to silence HRI and DELE1 in HEK293T and HeLa cells to exclude potential off-target effects of the previously used shRNA constructs. (Supplementary Fig. 1c and e)
- Repeated mitochondrial fractionation of cells treated with equivalent doses of CCCP and oligomycin to illustrate that DELE1-S and -VS formed after oligomycin were more retained in mitochondria fraction. (Fig. 5c)
- Included trans-resveratrol, an inhibitor of ATP synthase, as a trigger of MPIS and DELE1 cleavage in addition to oligomycin. (Supplementary Fig. 1b)
- Included OMA1 knockout 3T3 MEFs to demonstrate the baseline stress resulted from the removal of OMA1 and the potential challenge in the generation of CRISPR/Cas9-mediated OMA1 knockout cell lines. (Supplementary Fig. 4d and e)
- Added citations and adjusted order of figures to improve the flow of the manuscript.

b) Point-by-point response to reviewers' comments:

Reviewer #1: In 2020, two groups independently reported that DELE1 activates HRI-mediated integrated stress response (ISR) upon mitochondrial stress. In their model, DELE1 is cleaved by OMA1 upon mitochondrial stress, and cleaved DELE1 (DELE1-S) is released into the cytosol where it interacts with and activates cytosolic kinase HRI through its C-terminal TPR domains. In this manuscript, the authors found additional cleavage of DELE1 by HtrA2 presumably at DELE1 C-terminus (DELE1-VS). This newly identified DELE1 cleavage was induced specifically under mitochondrial import stress (MPIS), but not under mitochondrial stress that is accompanied with mitochondrial depolarization. The previous DELE1 detection system relied on its C-terminal HA tag, and thus it overlooked the C-terminal cleavage that was newly identified in this manuscript. Therefore, the main finding of this study (the HtrA2-mediated generation of DELE1-VS upon MPIS) is very interesting. However, it is not investigated whether the DELE1-VS generation plays critical roles in the MPIS-induced ISR activation. Also, some claims that show discrepancy with previous two Nature papers need more careful investigation, as these are relatively big claims. Particularly, regarding the possibility that DELE1-S is generated in the cytosol, more careful experimental design and data interpretation should be considered.

1. In Fig. 1, the authors mentioned the rapid turn-over of endogenous DELE1 protein. As this observation was already reported in the recent study (Fessler et al., Nature Commun., 2022), they should cite this paper in the body of manuscript.

We appreciate the recommendation and have included the citation in our revised manuscript.

2. In Fig. S2C, the authors claimed that DELE1 and HRI are not involved in the mitochondrial stress-induced ISR activation in HeLa cells. Although it looks their shRNA system efficiently suppressed both of HRI and DELE1 expression, I strongly recommend the authors to confirm their finding by siRNAs or even by creating CRISPR/Cas9-mediated knockout clones. Although the authors stated that HRI shRNA did not affect the endogenous DELE1 level in HEK293T cells (Fig. 1C), it seems HRI shRNA significantly reduced the endogenous DELE1 in HeLa cells (Fig. S2C), which is confusing, and raises several concerns, for example, several off-targets effects by these shRNAs. Any explanations for this?

We agree with the possibility that the single shRNA constructs for HRI and DELE1 used in the original manuscript to target HRI and DELE1 could have left open the possibility of off-targets effects. To address this issue, we included additional shRNA constructs that target HRI and DELE1 and demonstrated that the effects of HRI and DELE1 knockdown on the inhibition of ISR activation are consistent (Supplementary Fig. 1c-d). Regarding the processing of DELE1 following MPIS in HRI knockdown conditions, while in HEK293T cells the level of DELE1-S generated following CCCP is similar to the shRNA control (Supplementary Fig. 1c), in HeLa cells both shHRI constructs significantly reduced the amount of DELE1-S formed without significantly affecting the level of DELE1-L (Supplementary Fig. 1d-e), which confirms and reinforces our initial data, and shows that the effects observed were not due to off-target effects of the shRNAs. Currently we do not have a clear explanation for this observation in HeLa cells, which suggest the existence of a regulatory feedback loop involving HRI impacting on DELE1 in HeLa cells. We have also added a comment about this finding in the main text of the manuscript.

With regard to the generation of CRISPR/Cas9-mediated knockout clones, we understand the potential benefits of such system in theory. However, CRISPR/Cas9-mediated generation of KOs also has limitations, one of the main ones being the risk of long-term compensation effects in KO clones driven by the cell's need to adjust to the loss of a signaling pathway. While generating our shRNA knockdown cell lines, we noticed that shHRI and shDELE1 often exhibited stress after several weeks in culture (as a consequence we used a general practice to restart our shRNA

experiments freshly every few weeks to avoid selection of long-term effects), possibly because mitochondrial homeostasis is critical for cell's health in general. For these reasons, we were not enthusiastic about using KO experiments, especially because our knock-downs were overall pretty strong in terms of silencing the genes of interest.

3. In Fig. 1F, MG132 treatment clearly induces DELE1-VS, which is interesting. Although the authors interpreted this data as follows; "Thus, insufficient clearance of misfolded proteins by the proteasome system likely results in protein clogging at the site of protein import into mitochondria, thereby triggering the HRI-DELE1 axis of the ISR." However, we cannot rule out the possibility that DELE1-VS itself is rapidly degraded by proteasome. The authors should experimentally dissect these two possibilities.

The reviewer brings an interesting point. In order to answer this question, we first induced DELE1 cleavage by CCCP and oligomycin, and then studied the stability of DELE1-S and DELE1-VS following their formation by removing CCCP and oligomycin from the culture media and allowing the cells to recover (Supplementary Fig. 1f and g). A small dose of MG132 which by itself would not induce DELE1 cleavage was added to the recovery media to study the potential involvement of the proteasome in the degradation of DELE1 cleavage products. In CCCP recovery experiment, addition of MG132 led to elevated levels of DELE1-S and delayed recovery to baseline, thus strongly suggesting that the proteasome is indeed playing a role in the degradation of DELE1-S. In oligomycin recovery experiment, MG132 resulted in elevated levels of both DELE1-S and -VS, thus confirmed the results with CCCP and indicated the proteasome is indeed involved in the removal of both DELE1-S and -VS following MPIS. Thus, the reviewer is correct in speculating that one of the effects of MG132 might indeed be to allow for the stabilization and accumulation of homeostatically-produced DELE1-S and DELE1-VS.

However, we also obtained evidence that cytosolic accumulation of mitochondrial presequences can lead to DELE1 cleavage and the activation of ISR, which was demonstrated in cells with the over-expression of NLRX1 to mimic the inadequate removal of mitochondrial proteins by the proteasome (Fig. 1h and Supplementary Fig. 1h). Therefore, we feel that it remains difficult to fully assess what is the primary contributing factor of DELE1 cleavage following MG132 (stabilization of constitutively produced DELE1-S and DELE1-VS, or cytosolic accumulation of misfolded mitochondrial presequences). We therefore decided to include all the experiments described above in the revised manuscript and toned down the original interpretation by concluding that proteasomal inhibition by MG132 can lead to the accumulation of DELE1 cleavage products by both inducing their formation and preventing their degradation.

4. Related to Fig. 1F, MG132-induced ISR activation is actually abrogated by the deletion of DELE1 and HRI?

Previous study published from our lab¹ had indicated that ISR activation induced by MG132 was dependent on HRI. We confirmed this discovery in the following experiment (see panel below, not included in the manuscript; for reviewers only), and additionally showed that the ISR activation

induced by MG132 and another proteasome inhibitor (bortezomib) do not seem to be dependent on DELE1. This suggests that the induction of the ISR in response to proteasome inhibition is likely complex and, while it seems to depend mostly on HRI induction, the impact of the mitochondrial DELE1 axis is less crucial. This is consistent with our previous findings, which suggested that HRI serves as a general sensor of proteotoxicity in cells¹, so likely not only the proteotoxicity caused by MPIS.

5. In Fig. 3, the authors tried to estimate the actual cleavage site that generated DELE1-VS. However, their conclusion (DELE1-VS generated by both cleavage at N-terminus and C-terminus) is not fully supported by their data. As this is a very critical point, the actual cleavage site should be determined by other methods, for example, mass spec analysis.

We agree with the reviewer that finding the actual cleavage site of DELE1-VS is important. Since we had evidence that the cleavage of DELE1-VS was most likely to occur near the c-terminus end of DELE1, we generated a series of truncated mutants of DELE1 which each lacked 10 amino acids from the c-terminal end. Out of the mutants that we generated, where amino acids 456-466 were removed, we observed significant loss in the level of DELE1-VS following oligomycin treatment, without affecting the generation of DELE1-S (Fig. 3e). Interestingly, by aligning the peptide sequences of DELE1 from multiple species, amino acids 456-466 were found to be generally conserved only in mammals but not in other animal groups such as bird, reptile and fish (Fig. 3g). We then went on to mutate individual residues within the 10-amino acid long region in order to further narrow down to the residue that is responsible for the DELE1-VS cleavage. However, despite our efforts, unfortunately we could not identify any single amino acid substitution that could significantly abolish the DELE1-VS cleavage following oligomycin treatment (data not shown). Overall, these results suggested that the cleavage site of DELE-VS is likely promiscuous. The cleavage could be dependent on multiple amino acids within this region, and/or could be dependent more on the exposed peptide loop (rather than specific AAs) when the protein is stalled in the import machinery.

Finally, it is interesting to note that, in the original article that identify the cleavage site of DELE1-S to be histidine 142², the removal of amino acid 140-149 did not fully abolish the formation of DELE1-S (Fig. 3d of *Guo et al., Nature, 2020*), and up to this day, its exact AA sequence required for the generation of the “S” cleavage site is not clearly defined. This reinforces the notion that both DELE1-S and DELE1-VS cleavage sites could be partially promiscuous.

6. In Fig. 4, the authors stated that MPIS-induced ISR and mitophagy are independently induced. I agree with the importance of this observation, but it blocks the entire flow of the manuscript. Therefore, I recommend the authors to move these data to supplemental figures, and keep to a minimum description in the body of the manuscript.

We agree with the recommendation and have moved Fig. 4 to the supplementary data (now Supplementary Fig 3) in our revised manuscript.

7. In Fig. 5, the authors stated that OMA1 is not involved in DELE1 cleavage in HEK293T cells, but mediates the cleavage in HeLa cells. The authors only show the DELE1 cleaved form (S and VS) on their immunoblot in main Fig. 5 that addressed the OMA1 dependency in HEK293T cells, but they should also show the behavior of DELE1 full-length form. In Fig. S4A and S4B that addressed OMA1 dependency in HeLa cells, the authors showed the DELE1 full-length form. Although both DELE1-S and -VS forms reduced after OMA1 shRNA treatment, the recovery of the DELE1 full-length form was not observed under this condition, which is really confusing. Where is the un-cleaved DELE1 full-length form? This point should be carefully re-evaluated. Otherwise, the current

experimental setting and data interpretation are too premature to conclude the cell-type-dependent OMA1 dependency for DELE1 cleavage.

We agree that the questions regarding the detection and fate of full length DELE1 (DELE1-L) are important. In our hands, the detection of endogenous DELE1-L by Western blot has always been a bit difficult and somewhat variable since the start of using the endogenous DELE1 antibody, especially in HEK293T cells. We believe this is a complex issue that likely resulted from a combination of the short half-life and the low expression level of DELE1. To address the instability of DELE1-L, we included cells with shRNA that targets LONP1, which is a matrix protease that degrades unwanted proteins to maintain proteostasis, and was shown to degrade endogenously tagged DELE1 in a recently published study³. Interestingly, under baseline condition in HEK293T cells, DELE1-L became more detectable by our endogenous DELE1 antibody when the expression of LONP1 was silenced (Fig. 5a). Therefore, it is likely that in HEK293T cells, the bulk of DELE1-L is constantly imported and degraded in matrix, which leads to the low detectability at baseline. In HeLa cells, for an unknown reason, DELE1-L was relatively more detectable at baseline, as can be seen in the representative blots presented throughout the manuscript. We hypothesize that comparing to HEK293T cells, DELE1 could be expressed at higher level in HeLa cells, or the LONP1 protease could be less active in HeLa cells. Overall, the detectable DELE1-L is likely a net result of synthesis vs degradation, which could both be affected by multiple factors such as cell type, culture conditions and sensitivity of the batch of anti-DELE1 antibody used. Regarding the fate of un-cleaved DELE1-L in cells following MPIS, since we did not detect significant increase in DELE1-L levels in cells treated with MG-132 (Supplementary Fig. 1 e and f), the un-cleaved DELE1-L is likely not cleared by the proteasome system and the responsible mechanism remained unknown. Furthermore, rapid induction of ATF4 following MPIS could also attenuate the expression of novel DELE1 as a negative feedback mechanism to prevent the accumulation of DELE1 in the cytosol, which together with all the other mentioned factors, all likely contribute to the low detectable level of DELE1-L.

8. As mentioned in comment 2, the OMA1 dependency for DELE1 cleavage should be also confirmed by other gene suppression methods like siRNAs or CRISPR/Cas9-mediated gene editing.

We appreciate the recommendation by the reviewer regarding the siRNA and CRISPR/Cas9-mediated knockout. Similar to the comments about shHRI and shDELE1 cell lines mentioned above, in the original manuscript we have already included multiple shRNA constructs that targeted OMA1 to avoid potential off-target effect. In the revised manuscript we have now also included data from OMA1 knockout MEFs, and noticed the elevated level of stress in these cells when OMA1 is completely removed from the system, indicated by high baseline level of CHOP compared to WT cells (Supplementary Fig. 4d and e). This reinforced our impression that working with multiple shRNA constructs, rather than generating CRISPR-Cas9 KOs was probably the best approach for our study, likely because completely deleting an important homeostatic and housekeeping pathway such as the DELE1-HRI pathway in cells could be detrimental to cell's fitness.

9. In the body of manuscript on page 9, about the disappearance of OMA1 during CCCP treatment, the authors only mention YME1L involvement in OMA1 degradation. In addition, it is known that OMA1 degrades through self-cleavage mechanism. Therefore, following two reports also should be cited;

Baker, M.J., Lampe, P.A., Stojanovski, D., Korwitz, A., Anand, R., Tatsuta, T., and Langer, T. (2014). Stress-induced OMA1 activation and autocatalytic turnover regulate OPA1-dependent mitochondrial dynamics. EMBO J 33, 578-593. 10.1002/embj.201386474.

Zhang, K., Li, H., and Song, Z. (2014). Membrane depolarization activates the mitochondrial protease OMA1 by stimulating self-cleavage. *EMBO Rep* 15, 576-585. 10.1002/embr.201338240.

We agree with the recommendation and have included these citations in our revised manuscript.

10. In Fig. 6B, the authors argued that unlike O+A or CCCP, DELE1-S and -VS forms retained mitochondria fraction under Oligomycin-treated condition. However, the total amount of DELE1 is not equivalent among these stimuli in this figure. It looks more larger amount of DELE1-S and -VS forms are generated under Oligomycin treatment compared to O+A and CCCP stimuli. Thus, it is hard to interpret data as the authors did. In other figures, for example, in Fig 1B, DELE1 band shows similar intensity between CCCP and Oligomycin treatment on immunoblot. The authors repeat this experiment to confirm their conclusion.

We noticed the imbalance in the amount of DELE1 cleavage products formed after CCCP and oligomycin treatments, and understood this may interfere with the interpretation of the data. We have repeated these experiments and included the new panel in the revised manuscript (Fig. 5c).

11. In the body of manuscript on page 10 and Fig. 6F, the authors mentioned the possibility that DELE1-S form is not generated by OMA1 cleavage but it may be generated in the cytosol through unknown mechanisms. However, the current data sets and their data interpretation do not support their conclusion. First, their discussion (line 6-9, on page 10) is too speculative. Although DELE1-S form appears rapidly in the cytosol (Fig. 6A), the authors' interpretation of this data lacks scientific logic as no one tested the actual speed of retrotranslocation of mitochondrial proteins so far. Second, to support their conclusion experimentally, they examined whether DELE1 mutant that lacks "MTS" can be cleaved or not in Fig. 6C, but one important point should be clarified regarding experimental setting in this figure. Which region is deleted as "MTS" here? The authors should indicate the precise information of this mutant in the manuscript. It has been reported that DELE1 "MTS" has a unique feature – N-terminal 30 a.a. deletion does not prevent the mitochondrial localization of DELE1, and at least N-terminal 101 a.a. deletion is required for the efficient prevention of its mitochondrial localization (Fessler et al., Nature, 2020 and Guo et al., Nature, 2020). If the authors only deleted N-terminal small portion of DELE1 in this experiment, they should use DELE1 (d101) mutant.

We agree with the comment and have modified the discussion. Indeed, to test the speed of retrotranslocation of mitochondrial proteins would be complicated and we feel is out of the scope of this study. Regarding the Δ MTS DELE1 that we studied in this manuscript, we indeed used the DELE1 without the first 101 amino acids as our Δ MTS DELE1 and observed the cytosolic distribution of this mutant by immunofluorescence assay, which is consistent with the previous reports^{2,4} (Fig. 5e). We have included this information in the revised manuscript when we introduced the Δ MTS DELE1 mutant to avoid confusion.

12. As pointed out in comment 7, the authors should also examine whether the DELE1 full-length is recovered after AEBSF treatment in Fig. 7A and 7B. Otherwise, we cannot rule out the possibility that total expression of DELE1 is reduced by AEBSF for unknown reasons. This is also the case for HtrA2 knockdown in Fig. 7D and 7E. Of note, the recovery of the DELE1 full-length form was not observed under HtrA2 depleted condition in Fig. S5C, again which is confusing.

After carefully examining the blots in the original Fig. 7a, we agreed that it remained possible that high doses of AEBSF could reduce the total expression of DELE1 for unknown reasons. Potentially, high levels of AEBSF may lead to extensive inhibition of serine proteases and accumulation of proteotoxicity, which could eventually result in a general reduction in translation. The exact level of full length DELE1 in original Fig. 7a was also hard to determine as a result of reoccurring issue with detecting the full-length form of DELE1 in HEK293T cells that we have mentioned in comments above. We therefore decided to remove Fig. 7a to avoid confusion, which

contained the highest dose of AEBSF used in this manuscript, and instead kept the figure which shows a dose-curve of AEBSF (now Fig. 6b in the revised manuscript). For Fig. 7d in the original manuscript (Now Fig. 6d), we have updated the experiment with an additional shRNA construct targeting HtrA2, and all three HtrA2 constructs were found to not significantly affect baseline expression of full length DELE1 at baseline condition by Western blot. Regarding the lack of recovery and accumulation in DELE1-L in HeLa cells please refer to the response to comment 7 above.

13. In Fig. 7E, the authors also need to test the protease-dead mutant of HtrA2 in addition to reported PD-related HtrA2 mutants.

We agree with the reviewer that testing the catalytically-dead mutant of HtrA2 is important and can provide further evidence on its role in the cleavage of DELE1. We have generated the S306A mutant in which the catalytically active serine residue is changed to an alanine (S306A HtrA2)⁵ and tested this mutant in our cell line system (Fig. 6e). When WT HtrA2 and S306A HtrA2 were re-introduced to HtrA2 knockdown cells by over-expression, as expected, WT HtrA2 but not S306A HtrA2 was able to partially rescue the cleavage of DELE1-VS, thus further validating the importance of HtrA2 in the generation of DELE1-VS (Fig. 6e).

14. Again, the generation of DELE1-VS form under MPIS is really exciting finding. However, currently, the actual roles of this DELE1-VS form are totally unknown. The knockdown of HtrA2 attenuates the MPIS-induced ISR activation? Does DELE1-VS interact with HRI as DELE1-S does? The actual cleavage site in DELE1 C-terminus is not determined, but if the cleavage happens somewhere within C-terminal TPR domains, that cleavage may interrupt the proper folding TPR domains and thus it can disable the HRI binding. In that case, DELE1-VS form generation may have other meanings rather than ISR activation.

We agree with the reviewer that these questions regarding the potential physiological function of DELE1-VS are important. We first added the ATF4 blots in shHtrA2 cells challenged with MPIS inducers, and the induction of ATF4 was found to be partially attenuated when the expression of HtrA2 was silenced (Fig. 6d). Similarly, in HtrA2 knockout MEF cells, we observed the abolishment of ATF4 response when HtrA2 was removed from the system (Supplementary 5e and f). However, in HEK293T cells, HtrA2 knockdown resulted in both decreased levels of DELE1-VS and slight reduction in the amount of DELE1-S formed following MPIS, which could both contribute to the reduction in ATF4 response (Fig. 6d). In addition, we have yet to find a condition that only leads to the formation of DELE1-VS, and we actually doubt of the existence of such condition given that DELE-VS may only form after the cleavage of DELE1-S according to the model we proposed (see Fig. 5g). In order to study solely the function of DELE-VS, we investigated whether if DELE1-VS can interact with HRI by co-immunoprecipitation. In cells over-expressing with full length DELE1, after challenge of oligomycin, both DELE1-S and DELE1-VS were found to be interacting with HRI, which is consistent with the previous reports^{2,4} about the function of DELE1-S and suggested a similar function of DELE1-VS in the activation of HRI and ISR (Fig. 3h). Furthermore, since we showed the most likely cleavage site of DELE1-VS is within AA 456-466, compared to DELE1-S, DELE1-VS would lack the last TPR repeat which is located at AA 472-504^{2,4}. One of the previous reports² has tested a DELE1 mutant lacking the last TPR repeat (Δ C46aa DELE1) and demonstrated that the removal of the last TPR repeat did not abolish the ability of such DELE1 mutant to activate ATF4 reporter (please see Extended Data Fig. 3d of *Guo et al., Nature, 2020*), which is in support of the DELE1-VS-HRI interaction that we observed.

Reviewer #3 (Remarks to the Author):

In their manuscript titled "Cytosolic retention of the mitochondrial protease HtrA2 during mitochondrial protein import stress (MPIS) triggers the DELE1-HRI pathway," Bi et al. present that mitochondrial protein import stress is an overarching stress that triggers the DELE1-HRI-mediated integrated stress response. Using an antibody against DELE1, they detected distinct cleavage patterns of endogenous DELE1 depending on the types of MPIS. While only one short form of DELE1 is generated under depolarizing MPIS, under non-depolarizing MPIS such as the oligomycin condition, the authors uncovered an additional, even shorter form of DELE1 (DELE1-VS) generated by cleaving at the C-terminus of DELE1 by HtrA2 while stuck in the translocase pore upon import. While the study shows some important biology of endogenous DELE1, a critical protein that relays mitochondrial dysfunction to the ISR, further evidence is required to solidify the molecular mechanisms of DELE1-mediated ISR activation proposed in this paper.

More detailed comments:

1. This paper relies on an anti-DELE1 antibody (PA5-57712) to detect DELE1. However, according to the product information on the website (<https://www.thermofisher.com/antibody/product/DELE-Antibody-Polyclonal/PA5-57712>), this antibody gives rise to a strong signal in the nucleus in IHC experiments, suggesting strong non-specificity of this antibody. Although the antibody's specificity has been partially validated by shRNA knocking down of DELE1, it would be critical to further validate its specificity using a knockout (KO) cell line.

We appreciate the concern by the reviewer regarding the potential specificity issue with the DELE1 antibody that we used in this manuscript. Due to concern of non-specific target, this antibody was never used in immunofluorescence assay (Fig. 5 e and f). For IF studies, we instead relied on over-expression system of HA-tagged versions of DELE1 and studied the distribution by utilizing HA tag antibody. With regard to Western blot, we indeed found the existence of a few non-specific bands when using this antibody. However, we do observe the appearance of a specific band at around 40 kDa after challenge of MPIS inducers, and such band is consistent with the predicted molecular weight of DELE1-S and, importantly this band largely disappears in the two DELE1 shRNA knockdown cell lines (please see blots in Fig. 1 and Supplementary Fig. 1). We also observed a band at around 55 kDa in HeLa cells and sometimes in HEK293T cells that is at the similar predicted molecular weight of DELE1-L, which would again disappear in both DELE1 shRNA knockdown cell lines. Similarly, the novel DELE1-VS band at around 37 kDa also disappeared after DELE1 knockdown. Based on these results we concluded that these three bands are most likely representing true DELE1 species and we only assessed these three bands in the Western blot analysis of this manuscript. Regarding the generation of knockout cell lines, please see our response to the second comment of reviewer 1.

2. Using shRNA knockdown (KD) of important translocase components such as TOMM40, TOMM20, TOMM70, TIMM23, TIMM22, and MIA40 to trigger the MPIS, only knockdown of TIMM23 induces DELE1 cleavage and ATF4 activation. This contradicts previously reported data in Figure 6d and e (<https://doi.org/10.1038/s41467-022-29479-y>). TOMM40 is known as the main translocase pore, and its knockdown would block mitochondrial protein import. The data regarding MIA40 knockdown is not consistent with the MB-6 results. Therefore, the authors should provide an explanation and reconcile these observations.

We appreciate the questions raised by the reviewer regarding the potential inconsistency with a previous publication⁶. In the Figure 6d and e of the article mentioned, the knockdown of TOM40 also led to sharp decrease in the expression of TIM23 by Western blot, which could itself lead to the formation of DELE1-S and -VS. Therefore, the effect of TOM40 knockdown on the activation of ISR in their study could be explained by the secondary impact on TIM23. Indeed, in our

manuscript we did not observe the activation of ISR or the cleavage of DELE1 in siTOM40 cells, but the expression level of TIM23 was also not greatly affected in these cells (Fig. 2b and Supplementary Fig. 2). We also noticed the knockdown efficiency of our TOM40 siRNA was relatively low compared to other siRNA constructs that we used. We suspected that since TOM40 is an important importer on the outer membrane by forming the protein channel, cells with efficient knockdown of TOM40 would likely not survive the 2 to 4 days before harvest. We therefore decided not to try to identify a more efficient siRNA construct to repeat the experiment. Regarding the inconsistency between the activation of ISR by MIA40 knockdown and MB-6, we notice that the target of MB-6 is ERV1, which is another important protein in the MIA40-ERV1 pathway⁷, and targeting MIA40 alone may not be sufficient to induce the ISR activation and DELE1 induction that we observed with MB-6. Regardless, the main purpose of siRNA experiments is to illustrate the importance of TIM23, the key importer of DELE1 to the matrix, in the cleavage of DELE1 and the activation of ISR.

3. In Figure 5, the authors conclude that OMA1 is not required for DELE1 cleavage in HEK293 cells based on the results of shRNA knockdown of OMA1. Although the OMA1 knockdown cells showed a significant reduction in OMA1 protein levels, they still exhibited OPA1 cleavage with slower kinetics, as shown in Figure 5c. This suggests the presence of residual OMA1 activity, which might be sufficient to cleave DELE1. To achieve a clearer and more definitive interpretation, the authors should employ an OMA1 knockout (KO) cell line. This would provide better control and eliminate any potential residual activity of OMA1, leading to more robust conclusions.

This question is related to the comment No. 8 made by reviewer 1. Please see above for our response.

4. In Figure 6, the authors interpret the accumulation of DELE1-s in the cytosolic fraction upon CCCP treatment as a result of DELE1 cleavage in the cytosol. However, it is insufficient to conclude this, as it is possible that cleavage and release of DELE1 from mitochondria can occur simultaneously under CCCP treatment, while at a slower release rate under oligomycin treatment. Therefore, alternative explanations should be considered for the observed accumulation of DELE1-s in the cytosolic fraction upon CCCP treatment, and further experiments or analyses are needed to solidify the proposed mechanism.

We appreciate the questions raised by the reviewer regarding the rate of release of DELE1-S. For oligomycin treatment we observed similar kinetic of DELE1-S appearing in the cytosolic fraction compared to CCCP treatment (data not shown). Indeed, both cytosolic cleavage of DELE1 and the simultaneous cleavage and release of DELE1 from mitochondria can explain the accumulation of cytosolic DELE1-S. While we are not excluding the possibility of the latter, Δ MTS DELE1 (Δ AA 1-101) can still be cleaved into the DELE1-S despite its inability to be targeted to mitochondria (Fig. 5d), which made us favor the interpretation that the cleavage of DELE1-S would occur in the cytosol.

5. It is indeed very interesting that the DELE1-VS form is generated upon non-depolarizing MPIS. However, the physiological role of this fragment remains less clear. According to the model proposed in Figure 6, the cleavage of DELE1 at the C-terminus may be required for its release to the cytosol, where it is further cleaved at the "s" site to generate the very short form. However, this model does not explain why, under oligomycin treatment conditions, the majority of DELE1 exists as the short form. While Figure 7 demonstrates the role of HtrA2 in generating DELE1-VS, it would be important to investigate whether reducing the very short form of DELE1 abolishes the induction of ATF4 activation through ISR. Such experiments would provide valuable insights into the functional significance of the different DELE1 forms and their contributions to the integrated stress response.

The questions raised in the first part of this comment regarding the physiological function of DELE1-VS is related to the comment No. 14 made by reviewer 1. Please see above for our

response. With regard to why the majority of DELE1 cleavage products generated from oligomycin was DELE1-S, when oligomycin induces MPIS, we suspect the majority of DELE1-L are located in the cytosol awaiting import to mitochondria, while a small number of DELE1-L are in the process of import and are stuck in the import machinery (Fig. 5g). In our model, we proposed the purpose of VS cleavage is to free these stalled DELE1 to the cytosol, thus only the stalled DELE1 proteins would become DELE1-VS, while the majority of DELE1 are cleaved in the cytosol and become DELE1-S.

c) References

- 1 Abdel-Nour, M. *et al.* The heme-regulated inhibitor is a cytosolic sensor of protein misfolding that controls innate immune signaling. *Science* **365** (2019). <https://doi.org/10.1126/science.aaw4144>
- 2 Guo, X. *et al.* Mitochondrial stress is relayed to the cytosol by an OMA1-DELE1-HRI pathway. *Nature* **579**, 427-432 (2020). <https://doi.org/10.1038/s41586-020-2078-2>
- 3 Sekine, Y. *et al.* A mitochondrial iron-responsive pathway regulated by DELE1. *Mol Cell* **83**, 2059-2076.e2056 (2023). <https://doi.org/10.1016/j.molcel.2023.05.031>
- 4 Fessler, E. *et al.* A pathway coordinated by DELE1 relays mitochondrial stress to the cytosol. *Nature* **579**, 433-437 (2020). <https://doi.org/10.1038/s41586-020-2076-4>
- 5 Martins, L. M. *et al.* The serine protease Omi/HtrA2 regulates apoptosis by binding XIAP through a reaper-like motif. *J Biol Chem* **277**, 439-444 (2002). <https://doi.org/10.1074/jbc.M109784200>
- 6 Fessler, E., Krumwiede, L. & Jae, L. T. DELE1 tracks perturbed protein import and processing in human mitochondria. *Nat Commun* **13**, 1853 (2022). <https://doi.org/10.1038/s41467-022-29479-y>
- 7 Dabir, D. V. *et al.* A small molecule inhibitor of redox-regulated protein translocation into mitochondria. *Dev Cell* **25**, 81-92 (2013). <https://doi.org/10.1016/j.devcel.2013.03.006>

Reviewers' comments:

Reviewer #1 (Remarks to the Author):

Comments to the authors

In the revised manuscript, the authors well addressed my concerns. However, following two points should be reconsidered.

1. Now I agree with that the OMA1-dependency of the DELE1 cleavage (DELE1-S form generation) significantly differs depending on the cell line. However, it is still hard to accept the authors' conclusion that DELE1-S form is generated in the cytosol. The authors' data interpretation of Fig. 5B is logically immature. This concern was also pointed out by the other referee. I strongly suggest the authors to refrain from decisive conclusion on this point. Along with this, the authors should edit the text and the model in Fig. 5G appropriately in the current manuscript until the underlying detailed molecular mechanisms that support their hypothesis are fully uncovered in future study.

2. To address my and other referee's concern, the authors examined the HtrA2 involvement in the CCCP or Oligomycin-induced ISR activation, which is highly evaluated. However, following point is unclear. Although they argue that HtrA2 deletion impaired the ISR activation under both CCCP and Oligomycin-treated conditions, the effect is quite variable between HtrA2 shRNAs (Fig. 6D). Moreover, the CCCP-induced ATF4 accumulation was not suppressed at all in HtrA2 knockout MEF cells (Supplementary Fig. 5B), although I totally agree with that the Oligomycin-induced ATF4 accumulation was significantly suppressed in HtrA2 knockout MEF cells (Supplementary Fig. 5C). Through the entire manuscript, the authors emphasized that HtrA2-mediated DELE1-VS form generation is only induced by MPIS including Oligomycin, but not by CCCP. Therefore, to me, it really makes sense that HtrA2 is only involved in the Oligomycin-induced ISR activation, but not in the CCCP-induced one. But maybe the authors' thoughts are different? The authors should clarify this point.

Reviewer #3 (Remarks to the Author):

Bi. et. al conducted substantial additional experiments to strengthen the conclusions in the paper. I am pleased with their modifications of the manuscript and their responses to my initial review comments. These discoveries shed light on a potential link between the DELE1 pathway and Parkinson's disease. I strongly advocate for the publication of this article with only minor revision.

This paper raises the complexity of the ISR triggered by mitochondrial stress, both in cell-dependent as well as stress-dependent manner. Multiple previous studies showed that OMA1 is required to trigger the ISR upon mitochondrial stress, and both DELE1 and HRI are required to trigger ISR in Hela cells by CCCP and Oligomycin. I am still puzzled why the authors found the opposite. It could be the culture conditions which may affect metabolic state of these cell lines? I highly suggest the authors could discuss it as well as include more details in the method session on cell culture conditions.

DELE1 detection is heavily rely on epitope tagging. It is very encouraging that Bi. et al find an way to detect the endogenous DELE1. To benefit the community, it would be great for authors to include the DETAILED protocol for detection of DELE1 though WB like transfer method, blocking buffer, as well as their way of detection

Point-by-point response to reviewers' comments:

We would like to thank once again the reviewers and the editor for the time and effort spent on going through this revised manuscript and giving insightful comments. Please find below our point-by-point response to the last set of comments:

Reviewer #1 (Remarks to the Author):

Comments to the authors

In the revised manuscript, the authors well addressed my concerns. However, following two points should be reconsidered.

1. Now I agree with that the OMA1-dependency of the DELE1 cleavage (DELE1-S form generation) significantly differs depending on the cell line. However, it is still hard to accept the authors' conclusion that DELE1-S form is generated in the cytosol. The authors' data interpretation of Fig. 5B is logically immature. This concern was also pointed out by the other referee. I strongly suggest the authors to refrain from decisive conclusion on this point. Along with this, the authors should edit the text and the model in Fig. 5G appropriately in the current manuscript until the underlying detailed molecular mechanisms that support their hypothesis are fully uncovered in future study.

We thank the reviewer for this comment. Our conclusions that the “S” cleavage of DELE1 after CCCP treatment most likely occurs in the cytosol do not only stem from the Fig. 5b (cytosol/mitochondria fractionation), but also importantly from the data obtained using the DeltaMTS-DELE1 construct. Indeed, in Fig. 5d we observed that this construct, although lacking the mitochondria addressing sequence (which is further validated by our immunofluorescence data in Fig. 5e), can still be processed into DELE1-S following treatment with CCCP, which strongly argues for a cleavage occurring in the cytosol. We believe that the most likely explanation for these results is that the HTRA2 enzyme that can process DELE1 into DELE1-S may be retained into the cytosol following a defective import (as we observe in Fig. 6c), resulting in MPIS-induced cleavage of DeltaMTS-DELE1 to DELE1-S in the cytosol.

2. To address my and other referee's concern, the authors examined the HtrA2 involvement in the CCCP or Oligomycin-induced ISR activation, which is highly evaluated. However, following point is unclear. Although they argue that HtrA2 deletion impaired the ISR activation under both CCCP and Oligomycin-treated conditions, the effect is quite variable between HtrA2 shRNAs (Fig. 6D). Moreover, the CCCP-induced ATF4 accumulation was not suppressed at all in HtrA2 knockout MEF cells (Supplementary Fig. 5B), although I totally agree with that the Oligomycin-induced ATF4 accumulation was significantly suppressed in HtrA2 knockout MEF cells (Supplementary Fig. 5C). Through the entire manuscript, the authors emphasized that HtrA2-mediated DELE1-VS form generation is only induced by MPIS including Oligomycin, but not by CCCP. Therefore, to me, it really makes sense that HtrA2 is only involved in the Oligomycin-induced ISR activation, but not in the CCCP-induced one. But maybe the authors' thoughts are different? The authors should clarify this point.

We agree with the reviewer that the effect of HtrA2 silencing on DELE1 processing is more pronounced with regards to the VS fragment observed after oligomycin treatment, as compared to the S cleavage. However, we also do observe a reduction of the S form of DELE1 after CCCP treatment in shHtrA2 cells (see Fig. 6d). This effect is observed for the three shHtrA2 constructs that we tested (A, B and C), although we agree with the reviewer that the effect is more pronounced with shHtrA2 A, as we show on the representative WB in Fig. 6d. This is likely a consequence of the fact that this shRNA construct seemed to consistently silence HtrA2 expression better than the

other constructs (although we agree that this is not really obvious on this particular WB but we did observe this effect on the multiple repeats we performed).

With regards to the MEFs data, we agree with the reviewer's comment that the effect of HtrA2 knockout on CCCP-induced ATF4 induction was quite subtle in the repeat we presented in the manuscript (Supplementary Fig. 5b). As shown below, we have performed 4 independent repeats of this experiment and had initially opted for choosing Repeat #3 for the manuscript. Considering all these repeats, which all showed some degree of decrease in ATF4 and CHOP induction after CCCP treatment in the KO cells, we think it would be better to choose Repeat #1 instead, and thus decided to swap the panel in Supplementary Fig. 5b, which we think will provide more clarity in supporting further the notion that Htra2 indeed contributes to induction of the ISR in response to CCCP.

Reviewer #3 (Remarks to the Author):

Bi. et. al conducted substantial additional experiments to strengthen the conclusions in the paper. I am pleased with their modifications of the manuscript and their responses to my initial review comments. These discoveries shed light on a potential link between the DELE1 pathway and Parkinson's disease. I strongly advocate for the publication of this article with only minor revision.

This paper raises the complexity of the ISR triggered by mitochondrial stress, both in cell- dependent as well as stress-dependent manner. Multiple previous studies showed that OMA1 is required to trigger the ISR upon mitochondrial stress, and both DELE1 and HRI are required to trigger ISR in Hela cells by CCCP and Oligomycin. I am still puzzled why the authors found the opposite. It could be the culture conditions which may affect metabolic state of these cell lines? I highly suggest the authors could discuss it as well as include more details in the method session on cell culture conditions.

We thank the reviewer for the positive comments. We do believe that DELE1 cleavage is likely promiscuous (both S and VS cleavage seem to occur in disordered regions of the protein according

to AlphaFold prediction (see Fig. 3f)) and, therefore, we think that different enzymes (such as HtrA2 and OMA1) may contribute to DELE1 cleavage in cells, and the relative contribution of each enzyme may vary from cell line to cell line, thus explaining the partial discrepancy between our results and other studies. We also agree that other factors, such as metabolic differences in culture conditions or the type of serum used between different labs may be a contributing factor. As proposed by the reviewer we have included further details about cell culture condition in our Methods section.

DELE1 detection is heavily rely on epitope tagging. It is very encouraging that Bi. et al find an way to detect the endogenous DELE1. To benefit the community, it would be great for authors to include the DETAILED protocol for detection of DELE1 though WB like transfer method, blocking buffer, as well as their way of detection.

This is a good point. We have added further details about immunodetection of DELE1 in the Methods section.

Point-by-point response to reviewers' comments and summary of new data:

We would like to thank the reviewers and the editor for the time and effort spent on going through and giving insightful comments on how we could improve the manuscript. Please find below:

- a. The summary of the new data that we have included in the revised manuscript along with changes we have made.
- b. The point-by-point response to the reviewers' comments.

a) Summary of new data:

- Demonstrated the potential range of DELE1-VS cleavage site is most likely within AA 456-466 by generating DELE1 mutants with serial deletions in the c-terminus region and monitoring the formation of DELE1-VS following non-depolarizing MPIS. (Fig. 3e)
- Included co-immunoprecipitation assay to demonstrate the interaction of DELE1-S and VS with HRI. (Fig. 3h)
- Investigated the cleavage of DELE1-VS by WT HtrA2 and the catalytically-dead HtrA2 S306A mutant, to further confirm the role played by HtrA2 in the generation of DELE1-VS. (Fig. 6e)
- Included blots of ATF4 in HtrA2 knockdown cells and HtrA2 knockout 3T3 MEF cells treated with MPIS inducers to demonstrate the importance of HtrA2 in ISR activation following MPIS (Fig. 6d, Supplementary Fig. 5b-c)
- Included a recovery assay of CCCP and oligomycin with or without MG-132 to study the half-life of various DELE1 species, and investigated the potential involvement of proteasome in the degradation of DELE1. (Supplementary Fig. 1f-g)
- Included shRNA silencing of LONP1 to illustrate the importance of LONP1 during the degradation of full-length DELE1 that are imported to the matrix of mitochondria under baseline conditions. (Fig. 5a)
- Included additional shRNA constructs to silence HRI and DELE1 in HEK293T and HeLa cells to exclude potential off-target effects of the previously used shRNA constructs. (Supplementary Fig. 1c and e)
- Repeated mitochondrial fractionation of cells treated with equivalent doses of CCCP and oligomycin to illustrate that DELE1-S and -VS formed after oligomycin were more retained in mitochondria fraction. (Fig. 5c)
- Included trans-resveratrol, an inhibitor of ATP synthase, as a trigger of MPIS and DELE1 cleavage in addition to oligomycin. (Supplementary Fig. 1b)
- Included OMA1 knockout 3T3 MEFs to demonstrate the baseline stress resulted from the removal of OMA1 and the potential challenge in the generation of CRISPR/Cas9-mediated OMA1 knockout cell lines. (Supplementary Fig. 4d and e)
- Added citations and adjusted order of figures to improve the flow of the manuscript.

b) Point-by-point response to reviewers' comments:

Reviewer #1: In 2020, two groups independently reported that DELE1 activates HRI-mediated integrated stress response (ISR) upon mitochondrial stress. In their model, DELE1 is cleaved by OMA1 upon mitochondrial stress, and cleaved DELE1 (DELE1-S) is released into the cytosol where it interacts with and activates cytosolic kinase HRI through its C-terminal TPR domains. In this manuscript, the authors found additional cleavage of DELE1 by HtrA2 presumably at DELE1 C-terminus (DELE1-VS). This newly identified DELE1 cleavage was induced specifically under mitochondrial import stress (MPIS), but not under mitochondrial stress that is accompanied with mitochondrial depolarization. The previous DELE1 detection system relied on its C-terminal HA tag, and thus it overlooked the C-terminal cleavage that was newly identified in this manuscript. Therefore, the main finding of this study (the HtrA2-mediated generation of DELE1-VS upon MPIS) is very interesting. However, it is not investigated whether the DELE1-VS generation plays critical roles in the MPIS-induced ISR activation. Also, some claims that show discrepancy with previous two Nature papers need more careful investigation, as these are relatively big claims. Particularly, regarding the possibility that DELE1-S is generated in the cytosol, more careful experimental design and data interpretation should be considered.

1. In Fig. 1, the authors mentioned the rapid turn-over of endogenous DELE1 protein. As this observation was already reported in the recent study (Fessler et al., Nature Commun., 2022), they should site this paper in the body of manuscript.

We appreciate the recommendation and have included the citation in our revised manuscript.

2. In Fig. S2C, the authors claimed that DELE1 and HRI are not involved in the mitochondrial stress-induced ISR activation in HeLa cells. Although it looks their shRNA system efficiently suppressed both of HRI and DELE1 expression, I strongly recommend the authors to confirm their finding by siRNAs or even by creating CRISPR/Cas9-mediated knockout clones. Although the authors stated that HRI shRNA did not affect the endogenous DELE1 level in HEK293T cells (Fig. 1C), it seems HRI shRNA significantly reduced the endogenous DELE1 in HeLa cells (Fig. S2C), which is confusing, and raises several concerns, for example, several off-targets effects by these shRNAs. Any explanations for this?

We agree with the possibility that the single shRNA constructs for HRI and DELE1 used in the original manuscript to target HRI and DELE1 could have left open the possibility of off-targets effects. To address this issue, we included additional shRNA constructs that target HRI and DELE1 and demonstrated that the effects of HRI and DELE1 knockdown on the inhibition of ISR activation are consistent (Supplementary Fig. 1c-d). Regarding the processing of DELE1 following MPIS in HRI knockdown conditions, while in HEK293T cells the level of DELE1-S generated following CCCP is similar to the shRNA control (Supplementary Fig. 1c), in HeLa cells both shHRI constructs significantly reduced the amount of DELE1-S formed without significantly affecting the level of DELE1-L (Supplementary Fig. 1d-e), which confirms and reinforces our initial data, and shows that the effects observed were not due to off-target effects of the shRNAs. Currently we do not have a clear explanation for this observation in HeLa cells, which suggest the existence of a regulatory feedback loop involving HRI impacting on DELE1 in HeLa cells. We have also added a comment about this finding in the main text of the manuscript.

With regard to the generation of CRISPR/Cas9-mediated knockout clones, we understand the potential benefits of such system in theory. However, CRISPR/Cas9-mediated generation of KOs also has limitations, one of the main ones being the risk of long-term compensation effects in KO clones driven by the cell's need to adjust to the loss of a signaling pathway. While generating our shRNA knockdown cell lines, we noticed that shHRI and shDELE1 often exhibited stress after several weeks in culture (as a consequence we used a general practice to restart our shRNA

experiments freshly every few weeks to avoid selection of long-term effects), possibly because mitochondrial homeostasis is critical for cell's health in general. For these reasons, we were not enthusiastic about using KO experiments, especially because our knock-downs were overall pretty strong in terms of silencing the genes of interest.

3. In Fig. 1F, MG132 treatment clearly induces DELE1-VS, which is interesting. Although the authors interpreted this data as follows; "Thus, insufficient clearance of misfolded proteins by the proteasome system likely results in protein clogging at the site of protein import into mitochondria, thereby triggering the HRI-DELE1 axis of the ISR." However, we cannot rule out the possibility that DELE1-VS itself is rapidly degraded by proteasome. The authors should experimentally dissect these two possibilities.

The reviewer brings an interesting point. In order to answer this question, we first induced DELE1 cleavage by CCCP and oligomycin, and then studied the stability of DELE1-S and DELE1-VS following their formation by removing CCCP and oligomycin from the culture media and allowing the cells to recover (Supplementary Fig. 1f and g). A small dose of MG132 which by itself would not induce DELE1 cleavage was added to the recovery media to study the potential involvement of the proteasome in the degradation of DELE1 cleavage products. In CCCP recovery experiment, addition of MG132 led to elevated levels of DELE1-S and delayed recovery to baseline, thus strongly suggesting that the proteasome is indeed playing a role in the degradation of DELE1-S. In oligomycin recovery experiment, MG132 resulted in elevated levels of both DELE1-S and -VS, thus confirmed the results with CCCP and indicated the proteasome is indeed involved in the removal of both DELE1-S and -VS following MPIS. Thus, the reviewer is correct in speculating that one of the effects of MG132 might indeed be to allow for the stabilization and accumulation of homeostatically-produced DELE1-S and DELE1-VS.

However, we also obtained evidence that cytosolic accumulation of mitochondrial presequences can lead to DELE1 cleavage and the activation of ISR, which was demonstrated in cells with the over-expression of NLRX1 to mimic the inadequate removal of mitochondrial proteins by the proteasome (Fig. 1h and Supplementary Fig. 1h). Therefore, we feel that it remains difficult to fully assess what is the primary contributing factor of DELE1 cleavage following MG132 (stabilization of constitutively produced DELE1-S and DELE1-VS, or cytosolic accumulation of misfolded mitochondrial presequences). We therefore decided to include all the experiments described above in the revised manuscript and toned down the original interpretation by concluding that proteasomal inhibition by MG132 can lead to the accumulation of DELE1 cleavage products by both inducing their formation and preventing their degradation.

4. Related to Fig. 1F, MG132-induced ISR activation is actually abrogated by the deletion of DELE1 and HRI?

Previous study published from our lab¹ had indicated that ISR activation induced by MG132 was dependent on HRI. We confirmed this discovery in the following experiment (see panel below, not included in the manuscript; for reviewers only), and additionally showed that the ISR activation

induced by MG132 and another proteasome inhibitor (bortezomib) do not seem to be dependent on DELE1. This suggests that the induction of the ISR in response to proteasome inhibition is likely complex and, while it seems to depend mostly on HRI induction, the impact of the mitochondrial DELE1 axis is less crucial. This is consistent with our previous findings, which suggested that HRI serves as a general sensor of proteotoxicity in cells¹, so likely not only the proteotoxicity caused by MPIS.

5. In Fig. 3, the authors tried to estimate the actual cleavage site that generated DELE1-VS. However, their conclusion (DELE1-VS generated by both cleavage at N-terminus and C-terminus) is not fully supported by their data. As this is a very critical point, the actual cleavage site should be determined by other methods, for example, mass spec analysis.

We agree with the reviewer that finding the actual cleavage site of DELE1-VS is important. Since we had evidence that the cleavage of DELE1-VS was most likely to occur near the c-terminus end of DELE1, we generated a series of truncated mutants of DELE1 which each lacked 10 amino acids from the c-terminal end. Out of the mutants that we generated, where amino acids 456-466 were removed, we observed significant loss in the level of DELE1-VS following oligomycin treatment, without affecting the generation of DELE1-S (Fig. 3e). Interestingly, by aligning the peptide sequences of DELE1 from multiple species, amino acids 456-466 were found to be generally conserved only in mammals but not in other animal groups such as bird, reptile and fish (Fig. 3g). We then went on to mutate individual residues within the 10-amino acid long region in order to further narrow down to the residue that is responsible for the DELE1-VS cleavage. However, despite our efforts, unfortunately we could not identify any single amino acid substitution that could significantly abolish the DELE1-VS cleavage following oligomycin treatment (data not shown). Overall, these results suggested that the cleavage site of DELE-VS is likely promiscuous. The cleavage could be dependent on multiple amino acids within this region, and/or could be dependent more on the exposed peptide loop (rather than specific AAs) when the protein is stalled in the import machinery.

Finally, it is interesting to note that, in the original article that identify the cleavage site of DELE1-S to be histidine 142², the removal of amino acid 140-149 did not fully abolish the formation of DELE1-S (Fig. 3d of *Guo et al., Nature, 2020*), and up to this day, its exact AA sequence required for the generation of the “S” cleavage site is not clearly defined. This reinforces the notion that both DELE1-S and DELE1-VS cleavage sites could be partially promiscuous.

6. In Fig. 4, the authors stated that MPIS-induced ISR and mitophagy are independently induced. I agree with the importance of this observation, but it blocks the entire flow of the manuscript. Therefore, I recommend the authors to move these data to supplemental figures, and keep to a minimum description in the body of the manuscript.

We agree with the recommendation and have moved Fig. 4 to the supplementary data (now Supplementary Fig 3) in our revised manuscript.

7. In Fig. 5, the authors stated that OMA1 is not involved in DELE1 cleavage in HEK293T cells, but mediates the cleavage in HeLa cells. The authors only show the DELE1 cleaved form (S and VS) on their immunoblot in main Fig. 5 that addressed the OMA1 dependency in HEK293T cells, but they should also show the behavior of DELE1 full-length form. In Fig. S4A and S4B that addressed OMA1 dependency in HeLa cells, the authors showed the DELE1 full-length form. Although both DELE1-S and -VS forms reduced after OMA1 shRNA treatment, the recovery of the DELE1 full-length form was not observed under this condition, which is really confusing. Where is the un-cleaved DELE1 full-length form? This point should be carefully re-evaluated. Otherwise, the current

experimental setting and data interpretation are too premature to conclude the cell-type-dependent OMA1 dependency for DELE1 cleavage.

We agree that the questions regarding the detection and fate of full length DELE1 (DELE1-L) are important. In our hands, the detection of endogenous DELE1-L by Western blot has always been a bit difficult and somewhat variable since the start of using the endogenous DELE1 antibody, especially in HEK293T cells. We believe this is a complex issue that likely resulted from a combination of the short half-life and the low expression level of DELE1. To address the instability of DELE1-L, we included cells with shRNA that targets LONP1, which is a matrix protease that degrades unwanted proteins to maintain proteostasis, and was shown to degrade endogenously tagged DELE1 in a recently published study³. Interestingly, under baseline condition in HEK293T cells, DELE1-L became more detectable by our endogenous DELE1 antibody when the expression of LONP1 was silenced (Fig. 5a). Therefore, it is likely that in HEK293T cells, the bulk of DELE1-L is constantly imported and degraded in matrix, which leads to the low detectability at baseline. In HeLa cells, for an unknown reason, DELE1-L was relatively more detectable at baseline, as can be seen in the representative blots presented throughout the manuscript. We hypothesize that comparing to HEK293T cells, DELE1 could be expressed at higher level in HeLa cells, or the LONP1 protease could be less active in HeLa cells. Overall, the detectable DELE1-L is likely a net result of synthesis vs degradation, which could both be affected by multiple factors such as cell type, culture conditions and sensitivity of the batch of anti-DELE1 antibody used. Regarding the fate of un-cleaved DELE1-L in cells following MPIS, since we did not detect significant increase in DELE1-L levels in cells treated with MG-132 (Supplementary Fig. 1 e and f), the un-cleaved DELE1-L is likely not cleared by the proteasome system and the responsible mechanism remained unknown. Furthermore, rapid induction of ATF4 following MPIS could also attenuate the expression of novel DELE1 as a negative feedback mechanism to prevent the accumulation of DELE1 in the cytosol, which together with all the other mentioned factors, all likely contribute to the low detectable level of DELE1-L.

8. As mentioned in comment 2, the OMA1 dependency for DELE1 cleavage should be also confirmed by other gene suppression methods like siRNAs or CRISPR/Cas9-mediated gene editing.

We appreciate the recommendation by the reviewer regarding the siRNA and CRISPR/Cas9-mediated knockout. Similar to the comments about shHRI and shDELE1 cell lines mentioned above, in the original manuscript we have already included multiple shRNA constructs that targeted OMA1 to avoid potential off-target effect. In the revised manuscript we have now also included data from OMA1 knockout MEFs, and noticed the elevated level of stress in these cells when OMA1 is completely removed from the system, indicated by high baseline level of CHOP compared to WT cells (Supplementary Fig. 4d and e). This reinforced our impression that working with multiple shRNA constructs, rather than generating CRISPR-Cas9 KOs was probably the best approach for our study, likely because completely deleting an important homeostatic and housekeeping pathway such as the DELE1-HRI pathway in cells could be detrimental to cell's fitness.

9. In the body of manuscript on page 9, about the disappearance of OMA1 during CCCP treatment, the authors only mention YME1L involvement in OMA1 degradation. In addition, it is known that OMA1 degrades through self-cleavage mechanism. Therefore, following two reports also should be cited;

Baker, M.J., Lampe, P.A., Stojanovski, D., Korwitz, A., Anand, R., Tatsuta, T., and Langer, T. (2014). Stress-induced OMA1 activation and autocatalytic turnover regulate OPA1-dependent mitochondrial dynamics. EMBO J 33, 578-593. 10.1002/embj.201386474.

Zhang, K., Li, H., and Song, Z. (2014). Membrane depolarization activates the mitochondrial protease OMA1 by stimulating self-cleavage. *EMBO Rep* 15, 576-585. 10.1002/embr.201338240.

We agree with the recommendation and have included these citations in our revised manuscript.

10. In Fig. 6B, the authors argued that unlike O+A or CCCP, DELE1-S and -VS forms retained mitochondria fraction under Oligomycin-treated condition. However, the total amount of DELE1 is not equivalent among these stimuli in this figure. It looks more larger amount of DELE1-S and -VS forms are generated under Oligomycin treatment compared to O+A and CCCP stimuli. Thus, it is hard to interpret data as the authors did. In other figures, for example, in Fig 1B, DELE1 band shows similar intensity between CCCP and Oligomycin treatment on immunoblot. The authors repeat this experiment to confirm their conclusion.

We noticed the imbalance in the amount of DELE1 cleavage products formed after CCCP and oligomycin treatments, and understood this may interfere with the interpretation of the data. We have repeated these experiments and included the new panel in the revised manuscript (Fig. 5c).

11. In the body of manuscript on page 10 and Fig. 6F, the authors mentioned the possibility that DELE1-S form is not generated by OMA1 cleavage but it may be generated in the cytosol through unknown mechanisms. However, the current data sets and their data interpretation do not support their conclusion. First, their discussion (line 6-9, on page 10) is too speculative. Although DELE1-S form appears rapidly in the cytosol (Fig. 6A), the authors' interpretation of this data lacks scientific logic as no one tested the actual speed of retrotranslocation of mitochondrial proteins so far. Second, to support their conclusion experimentally, they examined whether DELE1 mutant that lacks "MTS" can be cleaved or not in Fig. 6C, but one important point should be clarified regarding experimental setting in this figure. Which region is deleted as "MTS" here? The authors should indicate the precise information of this mutant in the manuscript. It has been reported that DELE1 "MTS" has a unique feature – N-terminal 30 a.a. deletion does not prevent the mitochondrial localization of DELE1, and at least N-terminal 101 a.a. deletion is required for the efficient prevention of its mitochondrial localization (Fessler et al., Nature, 2020 and Guo et al., Nature, 2020). If the authors only deleted N-terminal small portion of DELE1 in this experiment, they should use DELE1 (d101) mutant.

We agree with the comment and have modified the discussion. Indeed, to test the speed of retrotranslocation of mitochondrial proteins would be complicated and we feel is out of the scope of this study. Regarding the Δ MTS DELE1 that we studied in this manuscript, we indeed used the DELE1 without the first 101 amino acids as our Δ MTS DELE1 and observed the cytosolic distribution of this mutant by immunofluorescence assay, which is consistent with the previous reports^{2,4} (Fig. 5e). We have included this information in the revised manuscript when we introduced the Δ MTS DELE1 mutant to avoid confusion.

12. As pointed out in comment 7, the authors should also examine whether the DELE1 full-length is recovered after AEBSF treatment in Fig. 7A and 7B. Otherwise, we cannot rule out the possibility that total expression of DELE1 is reduced by AEBSF for unknown reasons. This is also the case for HtrA2 knockdown in Fig. 7D and 7E. Of note, the recovery of the DELE1 full-length form was not observed under HtrA2 depleted condition in Fig. S5C, again which is confusing.

After carefully examining the blots in the original Fig. 7a, we agreed that it remained possible that high doses of AEBSF could reduce the total expression of DELE1 for unknown reasons. Potentially, high levels of AEBSF may lead to extensive inhibition of serine proteases and accumulation of proteotoxicity, which could eventually result in a general reduction in translation. The exact level of full length DELE1 in original Fig. 7a was also hard to determine as a result of reoccurring issue with detecting the full-length form of DELE1 in HEK293T cells that we have mentioned in comments above. We therefore decided to remove Fig. 7a to avoid confusion, which

contained the highest dose of AEBSF used in this manuscript, and instead kept the figure which shows a dose-curve of AEBSF (now Fig. 6b in the revised manuscript). For Fig. 7d in the original manuscript (Now Fig. 6d), we have updated the experiment with an additional shRNA construct targeting HtrA2, and all three HtrA2 constructs were found to not significantly affect baseline expression of full length DELE1 at baseline condition by Western blot. Regarding the lack of recovery and accumulation in DELE1-L in HeLa cells please refer to the response to comment 7 above.

13. In Fig. 7E, the authors also need to test the protease-dead mutant of HtrA2 in addition to reported PD-related HtrA2 mutants.

We agree with the reviewer that testing the catalytically-dead mutant of HtrA2 is important and can provide further evidence on its role in the cleavage of DELE1. We have generated the S306A mutant in which the catalytically active serine residue is changed to an alanine (S306A HtrA2)⁵ and tested this mutant in our cell line system (Fig. 6e). When WT HtrA2 and S306A HtrA2 were re-introduced to HtrA2 knockdown cells by over-expression, as expected, WT HtrA2 but not S306A HtrA2 was able to partially rescue the cleavage of DELE1-VS, thus further validating the importance of HtrA2 in the generation of DELE1-VS (Fig. 6e).

14. Again, the generation of DELE1-VS form under MPIS is really exciting finding. However, currently, the actual roles of this DELE1-VS form are totally unknown. The knockdown of HtrA2 attenuates the MPIS-induced ISR activation? Does DELE1-VS interact with HRI as DELE1-S does? The actual cleavage site in DELE1 C-terminus is not determined, but if the cleavage happens somewhere within C-terminal TPR domains, that cleavage may interrupt the proper folding TPR domains and thus it can disable the HRI binding. In that case, DELE1-VS form generation may have other meanings rather than ISR activation.

We agree with the reviewer that these questions regarding the potential physiological function of DELE1-VS are important. We first added the ATF4 blots in shHtrA2 cells challenged with MPIS inducers, and the induction of ATF4 was found to be partially attenuated when the expression of HtrA2 was silenced (Fig. 6d). Similarly, in HtrA2 knockout MEF cells, we observed the abolishment of ATF4 response when HtrA2 was removed from the system (Supplementary 5e and f). However, in HEK293T cells, HtrA2 knockdown resulted in both decreased levels of DELE1-VS and slight reduction in the amount of DELE1-S formed following MPIS, which could both contribute to the reduction in ATF4 response (Fig. 6d). In addition, we have yet to find a condition that only leads to the formation of DELE1-VS, and we actually doubt of the existence of such condition given that DELE-VS may only form after the cleavage of DELE1-S according to the model we proposed (see Fig. 5g). In order to study solely the function of DELE-VS, we investigated whether if DELE1-VS can interact with HRI by co-immunoprecipitation. In cells over-expressing with full length DELE1, after challenge of oligomycin, both DELE1-S and DELE1-VS were found to be interacting with HRI, which is consistent with the previous reports^{2,4} about the function of DELE1-S and suggested a similar function of DELE1-VS in the activation of HRI and ISR (Fig. 3h). Furthermore, since we showed the most likely cleavage site of DELE1-VS is within AA 456-466, compared to DELE1-S, DELE1-VS would lack the last TPR repeat which is located at AA 472-504^{2,4}. One of the previous reports² has tested a DELE1 mutant lacking the last TPR repeat (Δ C46aa DELE1) and demonstrated that the removal of the last TPR repeat did not abolish the ability of such DELE1 mutant to activate ATF4 reporter (please see Extended Data Fig. 3d of *Guo et al., Nature, 2020*), which is in support of the DELE1-VS-HRI interaction that we observed.

Reviewer #3 (Remarks to the Author):

In their manuscript titled "Cytosolic retention of the mitochondrial protease HtrA2 during mitochondrial protein import stress (MPIS) triggers the DELE1-HRI pathway," Bi et al. present that mitochondrial protein import stress is an overarching stress that triggers the DELE1-HRI-mediated integrated stress response. Using an antibody against DELE1, they detected distinct cleavage patterns of endogenous DELE1 depending on the types of MPIS. While only one short form of DELE1 is generated under depolarizing MPIS, under non-depolarizing MPIS such as the oligomycin condition, the authors uncovered an additional, even shorter form of DELE1 (DELE1-VS) generated by cleaving at the C-terminus of DELE1 by HtrA2 while stuck in the translocase pore upon import. While the study shows some important biology of endogenous DELE1, a critical protein that relays mitochondrial dysfunction to the ISR, further evidence is required to solidify the molecular mechanisms of DELE1-mediated ISR activation proposed in this paper.

More detailed comments:

1. This paper relies on an anti-DELE1 antibody (PA5-57712) to detect DELE1. However, according to the product information on the website (<https://www.thermofisher.com/antibody/product/DELE-Antibody-Polyclonal/PA5-57712>), this antibody gives rise to a strong signal in the nucleus in IHC experiments, suggesting strong non-specificity of this antibody. Although the antibody's specificity has been partially validated by shRNA knocking down of DELE1, it would be critical to further validate its specificity using a knockout (KO) cell line.

We appreciate the concern by the reviewer regarding the potential specificity issue with the DELE1 antibody that we used in this manuscript. Due to concern of non-specific target, this antibody was never used in immunofluorescence assay (Fig. 5 e and f). For IF studies, we instead relied on over-expression system of HA-tagged versions of DELE1 and studied the distribution by utilizing HA tag antibody. With regard to Western blot, we indeed found the existence of a few non-specific bands when using this antibody. However, we do observe the appearance of a specific band at around 40 kDa after challenge of MPIS inducers, and such band is consistent with the predicted molecular weight of DELE1-S and, importantly this band largely disappears in the two DELE1 shRNA knockdown cell lines (please see blots in Fig. 1 and Supplementary Fig. 1). We also observed a band at around 55 kDa in HeLa cells and sometimes in HEK293T cells that is at the similar predicted molecular weight of DELE1-L, which would again disappear in both DELE1 shRNA knockdown cell lines. Similarly, the novel DELE1-VS band at around 37 kDa also disappeared after DELE1 knockdown. Based on these results we concluded that these three bands are most likely representing true DELE1 species and we only assessed these three bands in the Western blot analysis of this manuscript. Regarding the generation of knockout cell lines, please see our response to the second comment of reviewer 1.

2. Using shRNA knockdown (KD) of important translocase components such as TOMM40, TOMM20, TOMM70, TIMM23, TIMM22, and MIA40 to trigger the MPIS, only knockdown of TIMM23 induces DELE1 cleavage and ATF4 activation. This contradicts previously reported data in Figure 6d and e (<https://doi.org/10.1038/s41467-022-29479-y>). TOMM40 is known as the main translocase pore, and its knockdown would block mitochondrial protein import. The data regarding MIA40 knockdown is not consistent with the MB-6 results. Therefore, the authors should provide an explanation and reconcile these observations.

We appreciate the questions raised by the reviewer regarding the potential inconsistency with a previous publication⁶. In the Figure 6d and e of the article mentioned, the knockdown of TOM40 also led to sharp decrease in the expression of TIM23 by Western blot, which could itself lead to the formation of DELE1-S and -VS. Therefore, the effect of TOM40 knockdown on the activation of ISR in their study could be explained by the secondary impact on TIM23. Indeed, in our

manuscript we did not observe the activation of ISR or the cleavage of DELE1 in siTOM40 cells, but the expression level of TIM23 was also not greatly affected in these cells (Fig. 2b and Supplementary Fig. 2). We also noticed the knockdown efficiency of our TOM40 siRNA was relatively low compared to other siRNA constructs that we used. We suspected that since TOM40 is an important importer on the outer membrane by forming the protein channel, cells with efficient knockdown of TOM40 would likely not survive the 2 to 4 days before harvest. We therefore decided not to try to identify a more efficient siRNA construct to repeat the experiment. Regarding the inconsistency between the activation of ISR by MIA40 knockdown and MB-6, we notice that the target of MB-6 is ERV1, which is another important protein in the MIA40-ERV1 pathway⁷, and targeting MIA40 alone may not be sufficient to induce the ISR activation and DELE1 induction that we observed with MB-6. Regardless, the main purpose of siRNA experiments is to illustrate the importance of TIM23, the key importer of DELE1 to the matrix, in the cleavage of DELE1 and the activation of ISR.

3. In Figure 5, the authors conclude that OMA1 is not required for DELE1 cleavage in HEK293 cells based on the results of shRNA knockdown of OMA1. Although the OMA1 knockdown cells showed a significant reduction in OMA1 protein levels, they still exhibited OPA1 cleavage with slower kinetics, as shown in Figure 5c. This suggests the presence of residual OMA1 activity, which might be sufficient to cleave DELE1. To achieve a clearer and more definitive interpretation, the authors should employ an OMA1 knockout (KO) cell line. This would provide better control and eliminate any potential residual activity of OMA1, leading to more robust conclusions.

This question is related to the comment No. 8 made by reviewer 1. Please see above for our response.

4. In Figure 6, the authors interpret the accumulation of DELE1-s in the cytosolic fraction upon CCCP treatment as a result of DELE1 cleavage in the cytosol. However, it is insufficient to conclude this, as it is possible that cleavage and release of DELE1 from mitochondria can occur simultaneously under CCCP treatment, while at a slower release rate under oligomycin treatment. Therefore, alternative explanations should be considered for the observed accumulation of DELE1-s in the cytosolic fraction upon CCCP treatment, and further experiments or analyses are needed to solidify the proposed mechanism.

We appreciate the questions raised by the reviewer regarding the rate of release of DELE1-S. For oligomycin treatment we observed similar kinetic of DELE1-S appearing in the cytosolic fraction compared to CCCP treatment (data not shown). Indeed, both cytosolic cleavage of DELE1 and the simultaneous cleavage and release of DELE1 from mitochondria can explain the accumulation of cytosolic DELE1-S. While we are not excluding the possibility of the latter, Δ MTS DELE1 (Δ AA 1-101) can still be cleaved into the DELE1-S despite its inability to be targeted to mitochondria (Fig. 5d), which made us favor the interpretation that the cleavage of DELE1-S would occur in the cytosol.

5. It is indeed very interesting that the DELE1-VS form is generated upon non-depolarizing MPIS. However, the physiological role of this fragment remains less clear. According to the model proposed in Figure 6, the cleavage of DELE1 at the C-terminus may be required for its release to the cytosol, where it is further cleaved at the "s" site to generate the very short form. However, this model does not explain why, under oligomycin treatment conditions, the majority of DELE1 exists as the short form. While Figure 7 demonstrates the role of HtrA2 in generating DELE1-VS, it would be important to investigate whether reducing the very short form of DELE1 abolishes the induction of ATF4 activation through ISR. Such experiments would provide valuable insights into the functional significance of the different DELE1 forms and their contributions to the integrated stress response.

The questions raised in the first part of this comment regarding the physiological function of DELE1-VS is related to the comment No. 14 made by reviewer 1. Please see above for our

response. With regard to why the majority of DELE1 cleavage products generated from oligomycin was DELE1-S, when oligomycin induces MPIS, we suspect the majority of DELE1-L are located in the cytosol awaiting import to mitochondria, while a small number of DELE1-L are in the process of import and are stuck in the import machinery (Fig. 5g). In our model, we proposed the purpose of VS cleavage is to free these stalled DELE1 to the cytosol, thus only the stalled DELE1 proteins would become DELE1-VS, while the majority of DELE1 are cleaved in the cytosol and become DELE1-S.

c) References

- 1 Abdel-Nour, M. *et al.* The heme-regulated inhibitor is a cytosolic sensor of protein misfolding that controls innate immune signaling. *Science* **365** (2019). <https://doi.org/10.1126/science.aaw4144>
- 2 Guo, X. *et al.* Mitochondrial stress is relayed to the cytosol by an OMA1-DELE1-HRI pathway. *Nature* **579**, 427-432 (2020). <https://doi.org/10.1038/s41586-020-2078-2>
- 3 Sekine, Y. *et al.* A mitochondrial iron-responsive pathway regulated by DELE1. *Mol Cell* **83**, 2059-2076.e2056 (2023). <https://doi.org/10.1016/j.molcel.2023.05.031>
- 4 Fessler, E. *et al.* A pathway coordinated by DELE1 relays mitochondrial stress to the cytosol. *Nature* **579**, 433-437 (2020). <https://doi.org/10.1038/s41586-020-2076-4>
- 5 Martins, L. M. *et al.* The serine protease Omi/HtrA2 regulates apoptosis by binding XIAP through a reaper-like motif. *J Biol Chem* **277**, 439-444 (2002). <https://doi.org/10.1074/jbc.M109784200>
- 6 Fessler, E., Krumwiede, L. & Jae, L. T. DELE1 tracks perturbed protein import and processing in human mitochondria. *Nat Commun* **13**, 1853 (2022). <https://doi.org/10.1038/s41467-022-29479-y>
- 7 Dabir, D. V. *et al.* A small molecule inhibitor of redox-regulated protein translocation into mitochondria. *Dev Cell* **25**, 81-92 (2013). <https://doi.org/10.1016/j.devcel.2013.03.006>